# Metacognitive efficiency in learned value-based choice

**Sara Ershadmanesh[1]\***, **Ali Gholamzadeh[1,2]**, **Kobe Desender[3]**, **Peter Dayan[1,2]**

**1** Department of Computational Neuroscience, MPI for Biological Cybernetics, Tuebingen, Germany,
**2** University of Tübingen, Tübingen, Germany, **3** Brain and Cognition, KU Leuven, Leuven, Belgium

\* sara.ershadmanesh@tuebingen.mpg.de

## Abstract

Metacognition, the ability to assess the quality of our own decisions, is a critical form of higher-order information processing. Metacognitive efficiency is therefore an essential measure of cognitive capability. When making decisions is simple, metacognitive judgments are also straightforward; thus, assessing metacognitive efficiency requires normalizing for the quality of underlying task performance. This is duly common in measures of efficiency popular in perceptual decision-making such as the M-ratio. However, such normalization is hard in reinforcement learning problems, because task difficulty changes dynamically. We therefore repurposed the central idea underlying the M-ratio, using confidence judgments to fashion a notional decision-maker (which we call a Backward model), assessing metacognitive sensitivity according to the quality of its virtual decisions, and quantifying metacognitive efficiency by comparing virtual and modelled actual qualities in the original decision-making task. We used simulated and empirical data to show that our measure of metacognitive sensitivity, the Backward performance, has comparable properties to other measures such as quadratic scoring, and that our measure of efficiency, the MetaRL.Ratio, is independent of empirical performance and is preserved across levels of task difficulty. We suggest that the MetaRL.Ratio as a promising tool for assessing metacognitive efficiency in value-based learning/decision-making.

## Author summary

When we make choices based on experience, we often judge how confident we are in them. The more accurate these so-called metacognitive judgments, the better, or more efficiently, we are able to evaluate our own decisions. Metacognitive efficiency plays an important role in many aspects of cognition and has been implicated in disease. In simple tasks such as spotting objects in a picture, many excellent ways have been developed to measure metacognitive efficiency. However, these methods struggle when the underlying difficulty of tasks fluctuates in

provided the original author and source are credited.

**Data availability statement:** Our code is publicly available at github.com/Ershadmanesh/meta-meassure. The data are accessible via github.com/kdesende/Two-armed-bandit-confidence.

**Funding:** This work was supported by Max Planck Society (SE, AG and PD), Humboldt Foundation (PD) and KU Leuven University (KD). The funders had no role in study design, data collection and analysis, decision to publish, or preparation of the manuscript.

**Competing interests:** The authors have declared that no competing interests exist.

incompletely known ways, as typically happens when people learn by trial and error. We introduce a new way to study metacognitive efficiency in such dynamic settings. We compare the synthetic performances of two models that can both solve the underlying task. A forward model is fit to match just the choices that participants made. A backward model is fit to match just the confidence judgements that accompanied those choices. The ratio between the two performances, the MetaRL.Ratio, is a measure of metacognitive efficiency that works well even as task difficulty shifts. In a game-like task in which reward values changed over time, the MetaRL.Ratio consistently measured metacognitive efficiency, unaffected by overall performance or confidence levels. Our measure helps extend the scope of metacognitive assessments.

## Introduction

Humans exploit their experiences to make good value-based decisions in changing environments. Along with their choices, they develop, and can express, degrees of confidence that those choices were correct or would be beneficial [1,2]. This form of higher order cognition is the most prevalent instance of metacognition, and is the topic of extensive computational, psychological and neurobiological investigations [3,4]. However, although there is a large body of work quantifying the metacognitive ability for the case of perceptual decision-making [5–11], an appropriate framework for estimating metacognitive ability in reinforcement learning tasks has yet to be developed.

In general, metacognition is characterized by three key quantities: bias, sensitivity and efficiency [12]. Bias is associated with the average of the levels that individuals use to report their confidence, with many individuals being under- or over-confident. Sensitivity quantifies the extent to which confidence reports about the rectitude of a decision distinguish between decisions that are actually correct or incorrect. Bias and sensitivity are partially distinguishable – thus, individuals may exhibit either underconfidence, despite high sensitivity, or overconfidence, with low sensitivity, at least provided that ceiling and floor effects are avoided. Metacognitive efficiency quantifies metacognitive ability. However, assessing efficiency is hard since the evaluation of sensitivity depends significantly on the quality of our decision-making (e.g., if our choices are always correct, then there is, by definition, no incorrect choice to distinguish). Therefore efficiency is traditionally evaluated by normalizing sensitivity according to a suitable measure of the performanc on the decisions underlying the confidence judgments.

Although many other approaches have been advanced [11,13–16], a popular way to assess metacognitive sensitivity in perceptual decision-making tasks is meta-$d'$ [12,17]. This starts from the observation that models of choice (as in signal detection theory; [18]) can automatically generate confidence values from their underlying decision variables, as the implied probability that the decision is correct [19]. In the terms used by [20], this defines a first-order model of confidence because all, and only, the

information underlying the choices (which are known as first-order decisions), is used for reporting confidence values. Thus, one can find the parameters of a choice model, notably the discriminability (duly called meta-$d'$), that make the distribution of its confidence values best match the empirical distribution of confidence reports of a participant. We call this fitted model the Backward model. Compared with other quantifications of metacognitive sensitivity such as the Quadratic Scoring Rule (QSR, a model-free measure defined in terms of the sum of the square quadratic difference between performance and reported confidence about choice across all trials; [21–24]) the value of meta-$d'$ is formally independent of any metacognitive bias (at least in the case that the signal detection theory model underlying meta-$d'$ is accurate [16]).

Meta-$d'$ measures metacognitive sensitivity, but, as noted above, it is also affected by empirical performance, i.e., the proportion of trials on which the participants chose the best option. For instance, it is very straightforward for a high-performing subject to report blindly high confidence about their usually correct choices, and yet seem metacognitively very sensitive. Thus, metacognitive efficiency measures based on meta-$d'$ normalize it according to the discriminability $d'$ of the original perceptual choice either divisively (creating the M-ratio) or subtractively [12]. Although the M-ratio is often less than 1, it can exceed this value if, for instance, participants can succeed at error monitoring. From this, the choices that, counterfactually, could have been made from the confidence judgments, would have been correct.

Unfortunately, the normalization for performance that is inherent to the M-ratio only works if task difficulty and performance are constant. The M-ratio is susceptible to inflation [25] when task difficulty varies. The latter case is ubiquitous in (reinforcement) learning tasks, such as two-arm bandits. In these tasks, difficulty varies at the outset, when participants are ignorant about the properties of the arms, and persistently, if the arms change qualities, being sometimes closer and sometimes further from each other. Thus, at present, we lack a valid approach to measure metacognitive efficiency in a dynamic learning context.

Here, we present an extension of the concept underlying the M-ratio to the learning domain. A picture of our framework is presented in Fig 1. We substitute a reinforcement learning model for the signal detection theory model inherent to the M-ratio. As in meta-$d'$, we fit the parameters of this model to the participants' confidence reports (the Backward model; paths "9" to "12" of Fig 1), and then assess its virtual performance, i.e., the proportion of trials on which the model chose the best option, on the underlying task, through simulation (paths 13 and 14). The Backward performance is our measure of metacognitive sensitivity. As in the M-ratio, we then normalize this virtual performance by the performance of an equivalent model fit to the actual choices of the participants (the Forward model; paths "1" to "4" of Fig 1). Finally, comparing performance of the Backward and Forward model leads to the MetaRL.Ratio, which quantifies metacognitive efficiency. Importantly, when fitting the Backward model to the confidence, we use an explicit method of scaling confidence judgments (Equation 9) to accommodate the participants' metacognitive biases.

We apply this process to data obtained from subjects making choices in a two-arm bandit task with reversals. We use a simple RL algorithm to characterize both the Forward and Backward models. We show that Backward performance is duly sensitive to the level of noise in confidence ratings, and, as anticipated, is appropriately related to the QSR. Also as intended, the MetaRL.Ratio was not significantly correlated with either empirical performance, or choice parameters, such as learning-rate or inverse temperature, as determined by fitting the Forward model. Appropriately, the MetaRL.Ratio was also unaffected by task difficulty, and, given our method of matching the model's confidence-ratings to empirical ones, it was less dependent on confidence bias (in the form of the empirical average confidence [26]). These results help establish the MetaRL.Ratio as the first model-based measure of metacognitive efficiency that can operate well in the domain of learning.

## Results

Our measure of metacognitive efficiency, the MetaRL.Ratio depends on fitting the parameters of one, Forward, RL model to subjects' choices; another, Backward, RL model to subjects' confidence ratings (assuming that these are generated as scaled versions of the model's choice probabilities); and then comparing the choice performance of these two models

PLOS Computational Biology

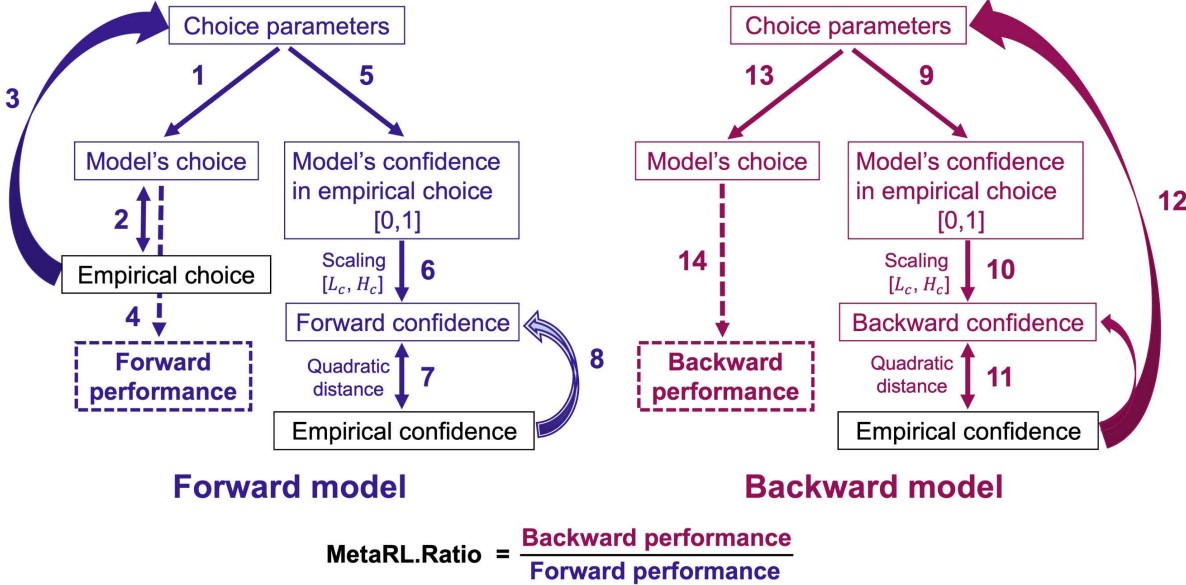

$$\text{MetaRL.Ratio} = \frac{\text{Backward performance}}{\text{Forward performance}}$$

**Fig 1. Framework of the MetaRL.Ratio based on forward and backward models.** Empirical choices (non-bold black) were characterized using an RL model (paths "1", "2" and "3") leading to choice parameters. The model's confidence in the empirical choices was determined based on its assessment of the probability that those choices were correct ("5"). These values were rescaled linearly ("6"; using lower and upper bounds $L_c$ and $H_c$ for subject $s$) to match the participant's empirical confidence reports by minimizing the squared error ("7", "8"). This produces what we call the Forward confidence. In a second comparison step, we also estimated the choice parameters through a Backward method by fitting empirical confidence ratings to the similarly scaled confidence of another RL model to the empirical choices ("9" to "12"). Finally, the behaviour of both the Forward and Backward models was simulated on new instances of the same task ("1", "13"), yielding estimates of the proportion of trials on which each model chose the best option, which we refer to as the Forward and Backward performance ("4", "14"). The Backward performance, which captures the influence of both empirical choices and confidence, serves as a measure of metacognitive sensitivity. To ensure a meaningful comparison, this measure was normalized by dividing it by Forward performance, resulting in the metacognitive efficiency measure known as the MetaRL.Ratio.

on new realizations of the task. We first describe the experiment used to illustrate the components of the MetaRL.Ratio, then show the results of fitting the Forward and Backward models, and thereby study the properties of the MetaRL.Ratio as a measure of metacognitive efficiency. Finally, we examine metacognitive efficiency and bias across different task conditions.

**Experiment.** 60 participants (54 after exclusions; see Methods) made choices between two slot machines, reporting their confidence in their selections on a scale of 1–5, and observing the reward associated with the chosen bandit (Fig 2). The rewards for the slot machines were generated from normal distributions, with mean values of 40 and 60, and variances of 8 in a low-variance condition and 16 in a high-variance condition (conducted within participants, but after a gap of $\frac{1}{2}$ to 3 days). There was also a subsequent test phase involving a common, medium-variance, condition which we do not analyze here. Participants knew that the average reward from the two options would reverse every 18–22 trials in an unsignalled manner, and they played 400 trials over 8 blocks in each condition of the task.

**Model of choice.** In order to enhance interpretability, our main analyses use a very simple reinforcement learning algorithm [27] for both the Forward and Backward models. This maintains separate values for each option, updating them according to the product of a learning-rate parameter, $\alpha$, and the prediction error (Equation 2; Materials and Methods), and choosing stochastically between the options based on a softmax with an inverse temperature parameter, $\beta$ (Equation 3; Materials and Methods). This parameter balances exploitation and undirected exploration. We also include lower ($L^{Fs}$, $L^{Bs}$) and upper ($H^{Fs}$, $H^{Bs}$) bounds for mapping the probability that the Forward ('F') and Backward ('B') models choose an

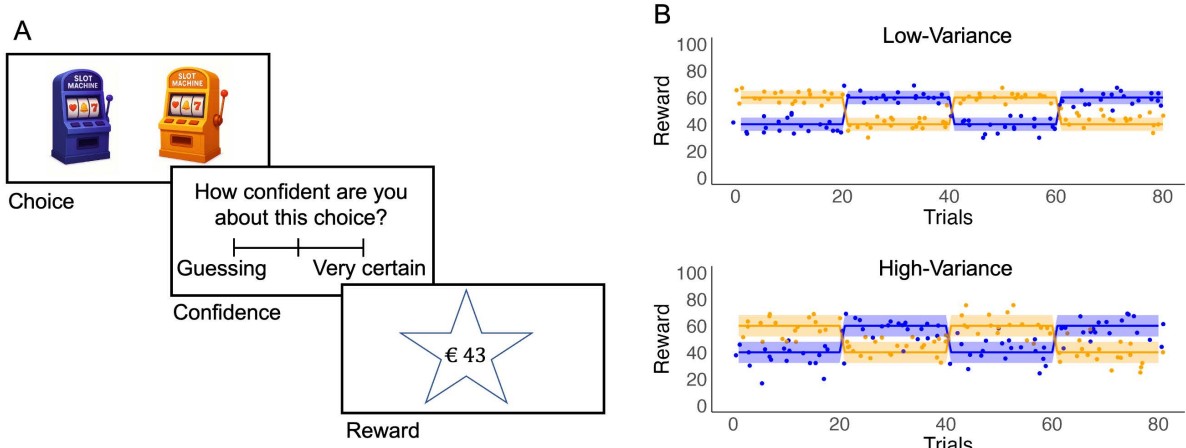

**Fig 2. Two-outcome reversal learning task. (A)** Participants made choices between two bandits (generated by an LLM: OpenAI ChatGPT, model o4-mini; see [OpenAI](# ) (2025)) and reported their confidence in the correctness of those choices (on a continuous scale) before receiving feedback on the outcome. **(B)** The bandits dispensed rewards, the dots, according to two normal distributions with means of 40 and 60, which alternated every $18-22$ trials. The variance for both options was set at 8 in a low-variance condition and set to 16 in a high-variance condition. Each participant completed both conditions in a counterbalanced order, with a time gap of $1/2-3$ days.

option into the reported confidence for subject $s$, on the grounds that individuals are likely to use the scales differently (see Equations 5; 9; Materials and Methods).

In S1 Text, we show that it is possible to recover the parameters of both models from simulated choice and confidence data (Fig A; B in S1 Text) (with more faithful recovery from the Backward model, since confidence values are more informative than binary choices), and that fitting the Backward model to noisy versions of simulated confidence judgments leads to parameters that generate a worse Backward performance (Fig C in S1 Text).

hlUnless otherwise stated, all the results in the main paper refer to the low-variance conditions. Results for the high-variance conditions are provided in Fig N-S in S2 Text.

**Comparison between Forward and Backward models.** We fit the Forward model to the choices of each subject $s$, determining the maximum likelihood values of the learning rate $\alpha^{Fs}$ and inverse temperature $\beta^{Fs}$. We then simulated new choices from the resulting Forward models on the same underlying task. Its performance, i.e., the proportion of trials on which the model chose the best option, was significantly lower than that of the empirical' choices (i.e., participants' observed choices) (M = .76 vs .86; Wilcoxon test: W = 7, p = 2.475e-10), as expected because we used the simplest RL model for interpretability (see Figs D to F in S1 Text for analogous metacognitive results from a more sophisticated RL algorithm and Figs G to K in S1 Text for results from model-based algorithms and their mixtures with model-free RL). We also fit lower $L^{Fs}$ and upper $H^{Fs}$ reporting bounds to minimize the misfit between the choice probability from the Forward model and the subjects' empirical confidence judgments.

We next found the values of the learning rate $\alpha^{Bs}$, inverse temperature $\beta^{Bs}$ and lower $L^{Bs}$ and upper $H^{Bs}$ reporting bounds of the Backward model by minimising the misfit—defined as the quadratic distance between the model's and empirical confidence ratings—to subject $s$'s confidence judgments. We then simulated new choices from the resulting Backward model on the same underlying task. The performance of the Backward model was significantly lower than that of the Forward model (Mean = .71 vs .76; Wilcoxon test: W = 326, p = 3.412e-04) and also than empirical choices (Mean = .71 vs .86; Wilcoxon test: W = 19, p = 4.81e-10) (Fig 3A). This was expected since the Backward model is fit to make good predictions of confidence ratings, not choices.

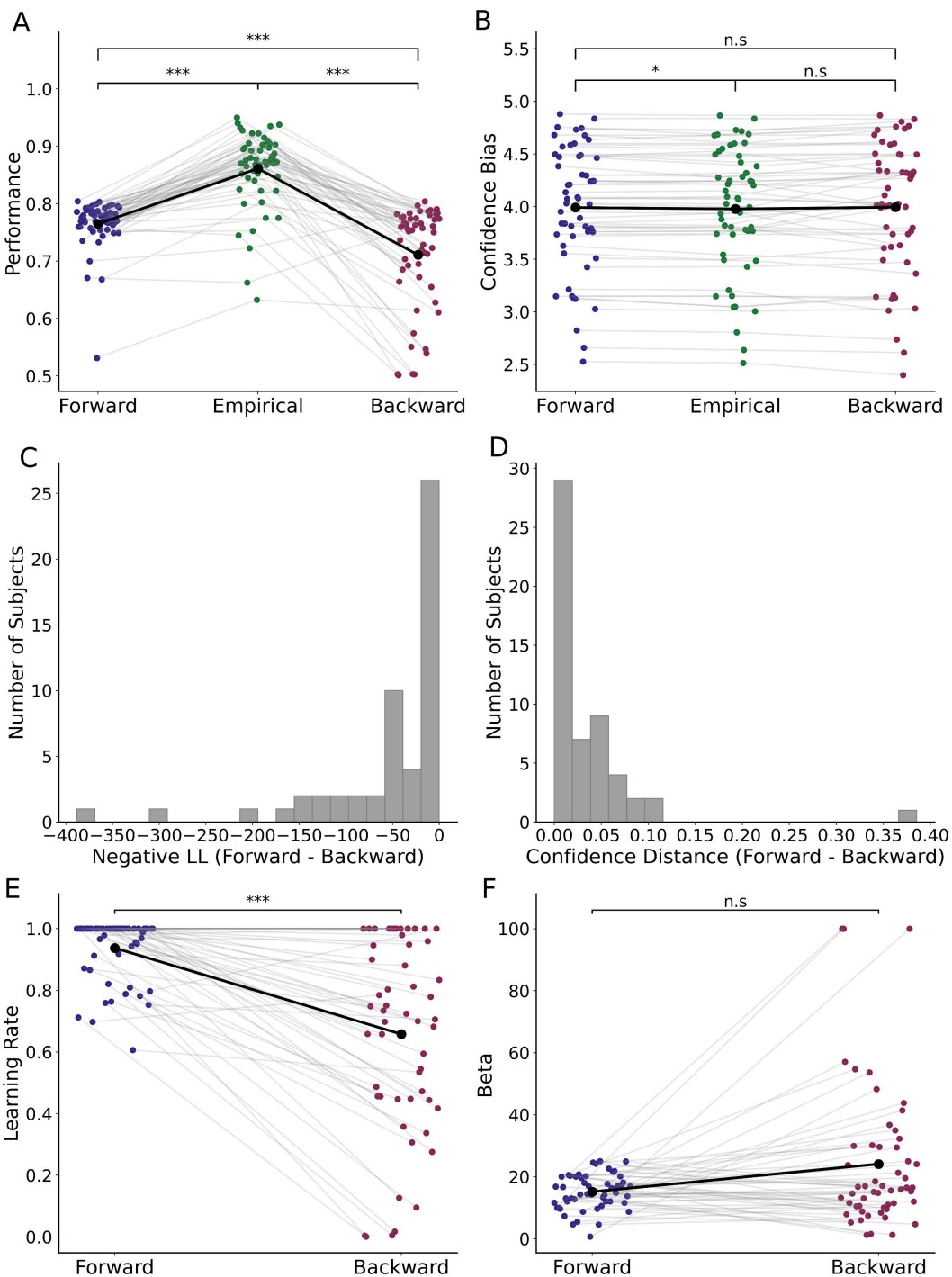

**Fig 3. Comparison between Forward and Backward models in choice, confidence and parameters. A)** The performance, proportion of trials on which each model chose the best option, of the Backward model was significantly lower than both empirical and Forward performance. Additionally, Forward performance significantly lagged behind empirical performance. **B)** The confidence bias levels of the Backward model were not significantly different from the Forward model and empirical data, while there was a weak but significant difference between the confidence bias of the Forward model and empirical data. **C)** The Forward model predicted choices better than the Backward model, as measured by the negative log likelihood. **D)** The confidence ratings of the Backward model were closer to the empirical data than those of the Forward model. **E)** The learning-rate was significantly lower in the Backward model compared to the Forward model. **F)** The inverse temperature parameter was not significantly different between two models. The dots in the plots represent the corresponding estimates for each subject in the low variance condition of the task.

There was no significant difference in confidence bias—quantified using the average confidence—between the (suitably scaled) Backward model and the empirical ratings (Mean = 3.99 vs. 3.98; Wilcoxon test: W = 890, p = .205), or between the Backward and Forward model (Mean = 3.99 vs. 3.99; Wilcoxon test: W = 691, p = .660) (Fig 3B). However, the confidence bias of the Forward model was marginally (though statistically significantly) higher than the confidence bias of the subjects (Mean = 3.99 vs 3.98; Wilcoxon test: W = 102, p = .017). The very small difference could come from our implementation of bias correction. To further quantify this, we compared the absolute differences between model-based and empirical confidence biases. The absolute difference between the Backward model and empirical ratings was significantly smaller than that between the Forward model and empirical ratings (Mean = 0.027 vs. 0.095; Wilcoxon test: W = 82, p = 1.29e-08).

The measure of fit of the Forward model is the negative log likelihood of the empirical choices. We can similarly evaluate the negative log likelihood of the predictions of the empirical choices of the Backward model, even though this was not how its parameter values were determined. As anticipated, the Forward model predicted choices better than the Backward model, as evidenced by the significantly lower negative log likelihood of empirical choices for the Forward model (Mean = 110.12 vs 163.58; Wilcoxon test: W = 0, p = 2.456e-10), for all of the subjects, the differences of negative log likelihood between Forward and Backward models were negative (Mean = -53.460, sd = 75.680) (Fig 3C). Conversely, the Backward model predicted confidence more proficiently, with a lower quadratic distance between its and the empirical rates than for the Forward model (Mean = .74 vs .78; Wilcoxon test: W = 1, p = 1.769e-10), for all of the subjects the confidence distance between Forward and Backward models were positive (Mean = .033, sd = .056) (Fig 3D).

We also compared the values of the fitted parameters between the models. The Backward model had a notably lower learning-rate than the Forward model (Mean = .66 vs .94; Wilcoxon test: W = 27, p = 4.875e-08) (Fig 3E). One potential source of this would be auto-correlation in confidence reports across trials (as in [28,29]). This arises because an effective method to produce auto-correlation in confidence reports from an RL model which lacks a direct mechanism for this, is to have more correlated choice $Q$-values. Such correlations can be achieved by having a lower learning rate. Thus, we checked the auto-correlation with lag one between empirical confidence-rates; there was considerable auto-correlation across subjects (Mean = .5056, SD = .2372). In line with the above hypothesis, the higher the auto-correlation of confidence-ratings, the lower the Backward learning-rate (low-variance; r = -.27, p = .049, high-variance; r = -.46, p = 4.68e-04) (Fig A in S2 Text). In addition, for the inverse temperature parameter, there was no significant difference between Forward and Backward models (Mean = 15.15 vs 24.11; Wilcoxon test: W = 954, p = .069) (Fig 3F); also the confidence lower and upper bound parameters ($L^{Fs}$, $L^{Bs}$; M = 2.35 vs 2.15; Wilcoxon test: W = 367.0, p = .023, $H^{Fs}$, $H^{Bs}$; M = 4.26 vs 4.51; Wilcoxon test: W = 353.0, p = 7.974e-04) (Fig B in S2 Text).

**Consistency with Quadratic Scoring Rule**. The QSR is a popular model-agnostic measure of metacognitive sensitivity [21–24]. The performance of the Backward model was correlated with the QSR (r = .50, p = 1.36e-04) (Fig 4A).

However, a prominent issue with QSR is that it depends on the confidence bias. This is evidenced by significant correlation between the two (r = .77, p = 1.13e-11). We therefore applied the same linear rescaling to the empirical confidence values (via $L_s^Q$ and $H_s^Q$) that would optimize the resulting QSR for subject $s$. We call the resulting metric the scaled-QSR. The scaled-QSR was correlated with confidence bias (r = .31, p = .022) but to a lesser degree than the conventional QSR (Z = 3.53, p = 4.10e-4) (see Fig C in S2 Text). Reassuringly, Backward performance was also correlated with Scaled-QSR (r = .39, p = .0031) (Fig 4B). Our measure of metacognitive sensitivity aligns with both QSR and scaled-QSR, demonstrating that Backward performance is consistent with estimates of metacognitive sensitivity, even after reducing dependence on confidence bias.

## Measure of metacognitive efficiency, MetaRL.Ratio

**Independence of the MetaRL.Ratio from choice parameters and confidence bias.** Measures of metacognitive efficiency need to take the effect of choice performance on metacognitive sensitivity into account [8,12]. We followed the idea underlying the M-Ratio by dividing the performance of the Backward model by that of the Forward model. We call this the MetaRL.Ratio.

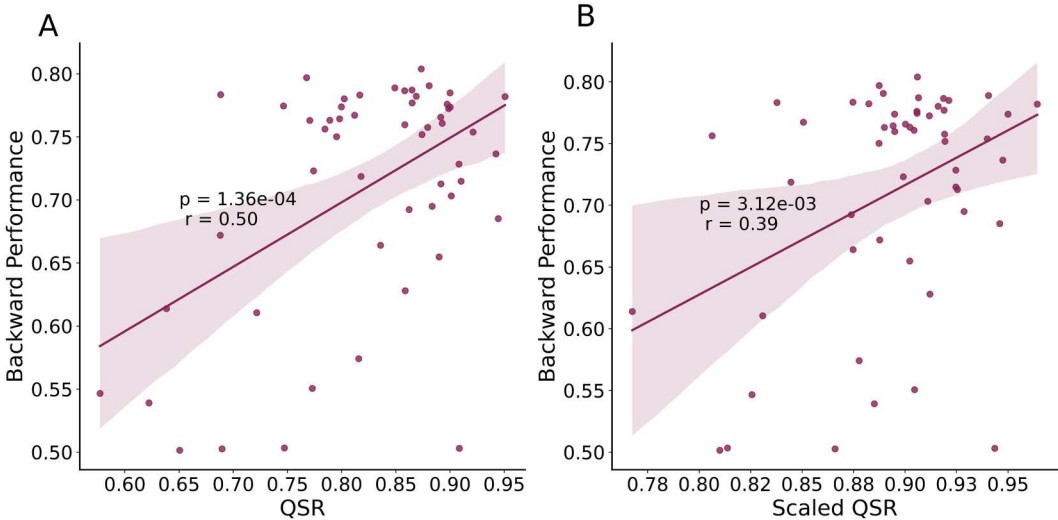

**Fig 4. Consistency with quadratic scoring rule, model-agnostic measure of metacognitive sensitivity. A)** Backward performance was significantly correlated with QSR. **B)** Backward performance was also significantly correlated with scaled-QSR, which determines a linear scaling of empirical confidence values to maximize QSR. The dots in the plots above represent the corresponding estimations for each subject. The dots in the plots show the estimates for each subject in the low variance condition of the task.

As anticipated, and even though the Forward model characterized choice imperfectly (Fig 3A), we observed no significant correlation between MetaRL.Ratio and the empirical performance (the proportion of trials on which participants chose the best option; r = -.07, p = .599) or indeed the learning-rate (r = .18, p = .193), or inverse temperature (p = .551, r = -.08, CI = [-.34,.19]) parameters of the Forward model (Fig 5). This is an important result, because a measure of metacognitive efficiency should ideally be independent of first-order task performance.

To investigate this independence more directly, we simulated choices and first-order confidence values using different configurations of the Forward model. Specifically, we set $\alpha^F$ to {.2, .5, .9}, and for each $\alpha^F$, we sampled $\beta^F$ at 10 equidistant points within the continuous range [5, 90]. This allowed us to assess the dependency of MetaRL.Ratio on $\beta^F$ for each $\alpha^F$. We similarly explored the dependency of MetaRL.Ratio on $\alpha^F$ for each $\beta^F$ by sampling $\beta^F$ values at {5, 15, 40} while varying $\alpha^F$ at 10 equidistant points within the continuous range [.05, 1]. The MetaRL.Ratio consistently hovered around a value of 1 as we varied the learning-rate (Mean = .999, sd = .007) and inverse temperature (Mean = 1, sd = .009), as expected for a first order model (Fig D in S2 Text). Thus, our measure of metacognitive efficiency remained independent of choice parameters, even in simulated behavior with free-ranging parameters.

If we estimated the MetaRL.Ratio on the human data without using our scaling method, it showed a strong correlation with confidence bias (r = .85, p = 2.01e-16). However, when including the scaling method in the MetaRL.Ratio calculation, the correlation with confidence bias (r = .40, p = .003) was less pronounced (Z = -4.304, p = 1.674e-05)(Fig 5B). Therefore, our linear scaling method considerably reduced the dependency of metacognitive efficiency on confidence bias, for both the model-agnostic measure of metacognitive sensitivity (scaled-QSR; as shown above) and the Backward/Forward model-based measure of metacognitive efficiency.

To test the possibility that part of the remaining dependence on the confidence bias arose from autocorrelation in confidence reporting (via an interaction with other estimated parameters), we incorporated a leaky representation of confidence [29,30] into our Backward model, such that confidence was predicted as a linear combination of the Backward confidence from the current trial and the reported confidence on the previous trial (see Equation 28; Materials and Methods). Using this leaky reporting mechanism, the Backward model fit empirical choices and confidence ratings

PLOS Computational Biology

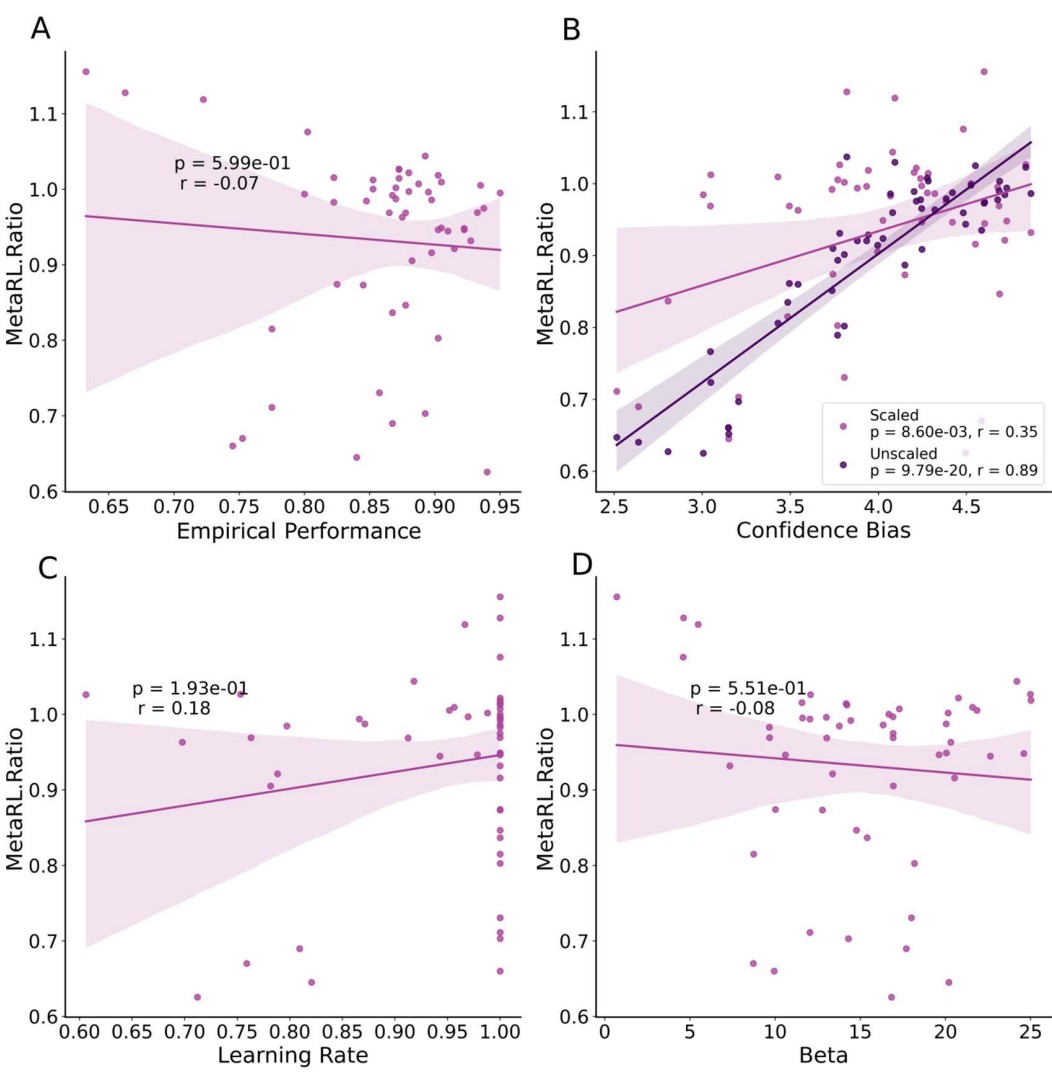

**Fig 5. Relationship between MetaRL.Ratio and empirical choice parameters. A)** The MetaRL.Ratio, our measure of metacognitive efficiency, was independent of empirical performance (the proportion of trials on which participants chose the best option). **B)** The correlation between MetaRL.Ratio and confidence bias decreased after applying the confidence scaling method. **C & D)** The MetaRL.Ratio was not significantly correlated with the inverse temperature (C) or the learning-rate (D) of the Forward model. The dots in the plots show the estimates for each subject in the low variance condition of the task.

better than our original Backward model (Mean = .73 vs .74, W = 10, p = 4.22e-10). Furthermore, the MetaRL.Ratio, when updated with the leaky confidence representation, was no longer dependent on confidence bias (r = .19, p = .161) (Fig E in S2 Text).

As we reported earlier, the learning rate of the Backward model was lower than that of the Forward model and lower learning rate of the Backward model was associated with higher auto-correlation of confidence-rating. As expected, the learning rate of the Leaky Backward model was higher than that of the Backward model (Mean = .71 vs .66, W = 98, p = 9.868e-06) while lower than the Forward model (Mean = .71 vs .94, W = 49, p = 4.836e-07) (Fig F in S2 Text). We refer to Fig 12 in S2 Text for the correlation between the Leaky Backward learning rate and the autocorrelation of confidence ratings, as well as the comparison with the Backward model.

**Independence of MetaRL.Ratio from task difficulty.** We noted that empirical performance and metacognitive sensitivity are typically correlated. This was also evident in our data when the task was made more difficult (in the high variance condition), with empirical (Mean = .86 vs .78; Wilcoxon test: W = 125, p = 1.081e-07), Forward (Mean = .76 vs .68; Wilcoxon test: W = 57, p = 3.679e-09) and Backward performance (Mean = .71 vs .64; Wilcoxon test: W = 207, p = 4.096e-06) all decreasing as both models were impacted by empirical choices (Fig 6A). However, consistent with its intended role as a measure of metacognitive efficiency, the MetaRL.Ratio was not statistically significantly different (Mean = .93 vs .95; Wilcoxon test: W = 858, p = .322) (Fig 6B).

The confidence bias was lower in the high- than the low-variance condition for empirical (Mean = 3.79 vs 3.98; Wilcoxon test: W = 400, p = .003), Forward (Mean = 3.99 vs 3.83; Wilcoxon test: W = 443, p = .01), and Backward (Mean = 3.99 vs 3.80; Wilcoxon test: W = 438, p = .009) models (Fig 6C). Thus, confidence bias was lower in the more difficult task condition, and this effect was replicated by both the Forward and Backward models, which were influenced by empirical choices.

Thus, our measure of metacognitive efficiency provides the opportunity to distinguish the effects of task difficulty on two aspects of metacognition: efficiency and bias. While the former remains independent of task difficulty, the latter is dependent on it.

The Forward learning-rate, $\alpha^{Fs}$, was slightly lower in the more difficult high-variance condition (Mean = .89 vs .94, W = 288, p = .041), while the Backward learning-rate, $\alpha^{Bs}$, did not significantly differ (Mean = .63 vs .66, W = 655, p = .76) (Fig 7A). In both the Forward and Backward models, the inverse temperature was significantly lower in the high-variance condition implying more stochastic choices (Forward model, $\beta^{Fs}$: M = 10.23 vs 15.15, W = 166, p = 7.07e-07; Backward

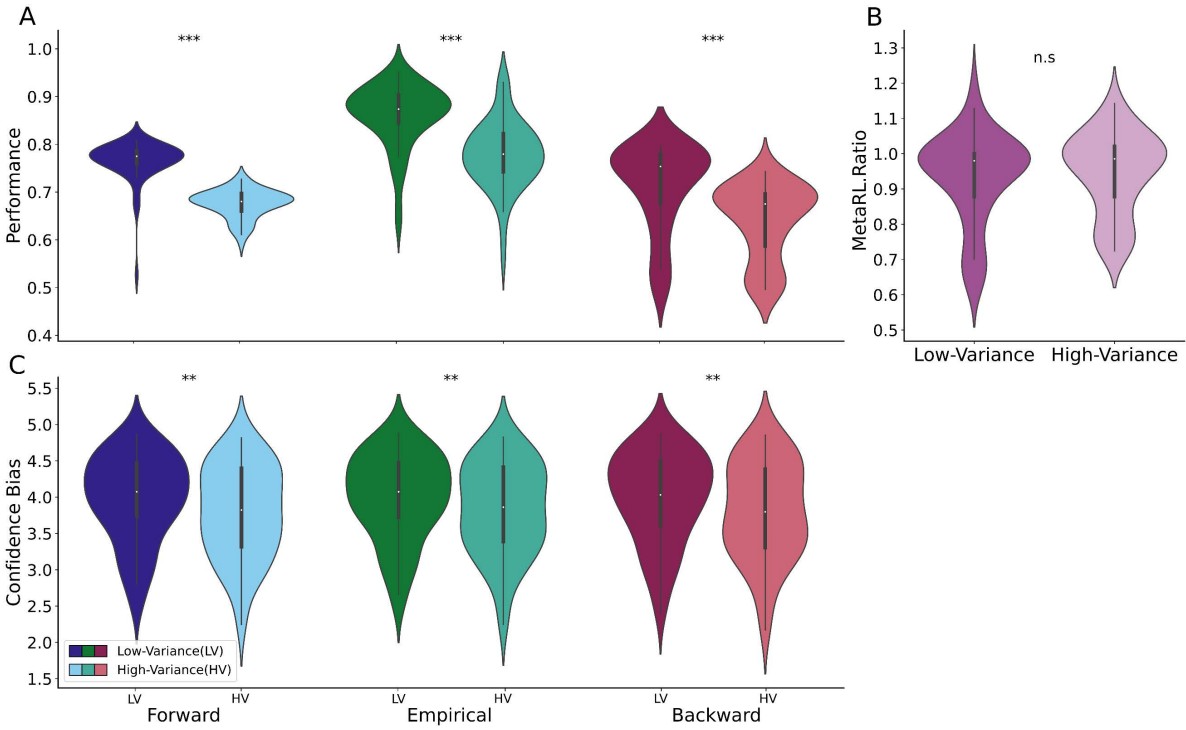

**Fig 6. Independence of MetaRL.Ratio from task difficulty. A)** Empirical performance (green), Forward performance (blue) and Backward performance (red) were significantly lower in the high- than the low-variance condition (HV versus LV). **B)** The MetaRL.Ratio was not significantly different between the two levels of task difficulty. **C)** The confidence bias was significantly lower in high-variance rather than low-variance condition of task according to empirical data (green), Forward model (blue) and Backward model (red).

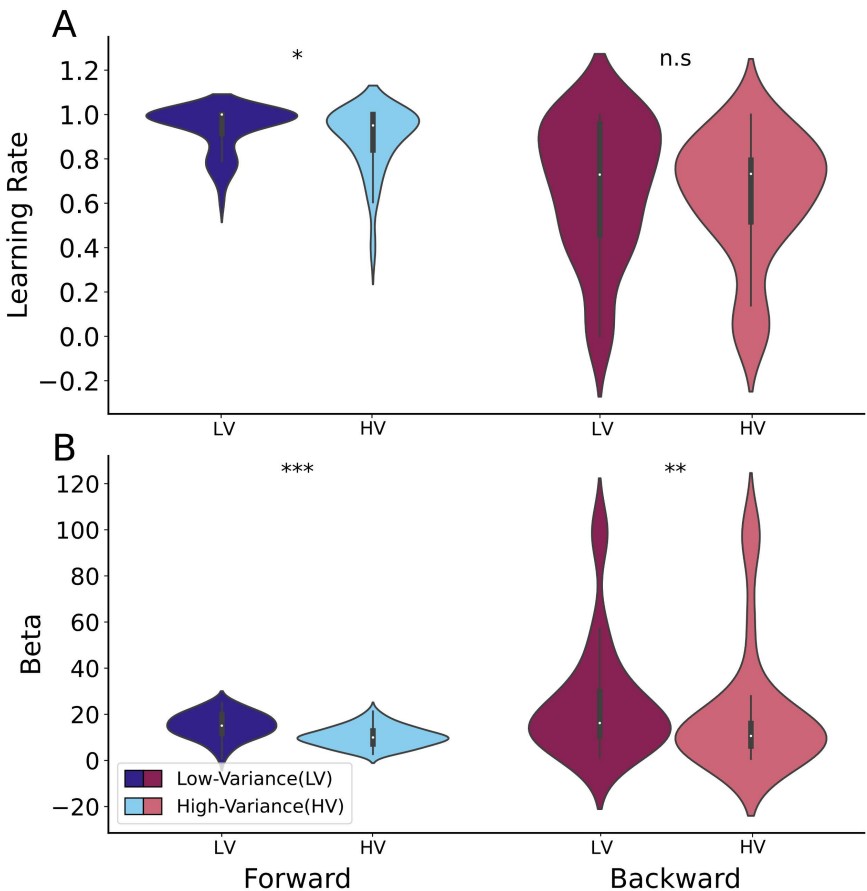

**Fig 7. Parameter comparison between two levels of task difficulty. A)** The learning-rate of the Forward model was slightly lower in the high-variance (HV) condition than the low-variance (LV) one of the task, while the difference was not significant for the Backward model. **B)** The inverse temperature was significantly lower in high versus low difficulty, strongly for the Forward model, but modestly so for the Backward model. The dots shown in the above plots correspond to the estimates for each subject.

model, $\beta^{Bs}$: M = 20.13 vs 20.68, W = 423, p = .01) (Fig 7B). The influence of task conditions on confidence bound parameters were shown at Fig G in S2 Text.

We then looked at the correlation across individuals between empirical performance and confidence bias for the two task conditions. Somewhat unexpectedly, neither empirical performance (r = .07, p = .621), nor the performance of the Forward model was significantly correlated between the two levels of task difficulty (r = -.05, p = .747). However, the performance of the Backward model (r = .39, p = .003) and the MetaRL.Ratio, were significantly, but slightly, correlated between the two levels of difficulty (r = .35, p = .011) (Table 1). By comparison, confidence bias values were strongly correlated between the two levels of difficulty, Empirical (confidence bias; r = .70, p = 4.86e-9), Forward (confidence bias; r = .70, p = 4.16e-9) and Backward (confidence bias; r = .65, p = 9.42e-08) (Table 2 for confidence bias). The correlation in Backward confidence bias between the two conditions was stronger than the corresponding correlation in MetaRL.Ratio (Z = -2.412, p = .015). This aligns with findings by [26], who observed across various domains that metacognitive bias more consistently maintains the ranking of subjects across different difficulty levels than metacognitive efficiency.

**Assessment Using a Refined Model-Free Agent.** As noted, we used a very simple, two-parameter, RL model for ease of interpretation, but at the expense of model fit. We therefore adapted this model to include separate learning rates

**Table 1. Correlation of performance and MetaRL.Ratio between Low- and High-Variance Conditions. Empirical and Forward model performance were not correlated between the two task conditions. In contrast, both the Backward performance (red) and MetaRL.Ratio (purple) were.**

| Performance | Empirical | Forward | Backward | MetaRL.Ratio |
|---|---|---|---|---|
| Correlation | .07 | -.05 | .39 | .35 |
| P-value | .621 | .747 | .003 | .011 |

**Table 2. Correlation of confidence bias between Low- and High-Variance Conditions. The confidence bias was correlated between two task difficulties for empirical data and also Forward and Backward models.**

| confidence bias | Empirical | Forward | Backward |
|---|---|---|---|
| Correlation | .70 | .70 | .65 |
| P-value | 4.86e-9 | 4.16e-9 | 9.42e-08 |

for positive and negative prediction errors, and a single (typically negative) learning rate for counterfactual updating of the unchosen option (given that subjects knew that the bandits alternated in quality). As a Forward model, this fit the subjects' choices more proficiently (BIC = 199.19 vs 244.20, W = 1.0, p = 1.720e-10) but still performed worse than the participants themselves on the task (Mean = .79 vs .86; Wilcoxon test: W = 118, p = 7.57e-08) (see Fig D in S1 Text). Nevertheless the advantages of our metacognitive measures remained consistent when using them for both Forward and Backward models (see Fig E; F in S1 Text).

**Comparison with a Hybrid Model of Choice.** We have so far only used model-free (MF) reinforcement learning (RL) methods. Since the performance of these models was still lower than that of the participants, we considered the possibility that there was an influence of the fact that the participants were told that reversals would occur every 18–22 trials. To try to capture human behavior more proficiently, we therefore applied a Hidden Semi-Markov Model (which we refer to as a form of model-based (MB) RL; see Equations 18–26; Materials and Methods) and a Mixture Model, where choice probabilities combined MF and MB action values (Equation 27; Materials and Methods). As a Forward model, MB RL fit the choices of participants was less well supported than MF RL (AIC = 295.44 vs. 224.23, W = 260, $p_{Bonf}$ = 9.783e-05 (Bonferroni-corrected across three tests); BIC = 303.42 vs. 232.22, W = 260, $p_{Bonf}$ = 9.783e-05). By contrast, the fit of the Mixture model was better than that of MF RL using AIC to penalize complexity (AIC = 193.26 vs. 224.23, W = 93, $p_{Bonf}$ = 6.72e-08). The Mixture model also fit better based on BIC, but this difference was not significant (BIC = 213.22 vs. 232.22, W = 93, $p_{Bonf}$ = 1.00). In addition, the Mixture model provided a better fit to participants' choices than MB RL according to both measures. (AIC = 193.26 vs. 295.44, W = 1, $p_{Bonf}$ = 5.16e-10; BIC = 213.22 vs. 303.42, W = 5, $p_{Bonf}$ = 6.459e-10) (see Fig G in S1 Text). Although the Mixture model's performance was marginally further from empirical' performance than MB RL (Fig H in S1 Text), overall it was the best-fitting model of empirical' choices.

In both the MB and Mixture models, participants' prior beliefs about when reversals occur were modeled as a negative binomial distribution over block duration, summarized by its mean, $\bar{t}_{rev}$, (Equations 22, Materials and Methods; see Fig H;I in S2 Text). Intuitively, larger values of the mean correspond to expecting reversals to be less frequent (i.e., longer stable blocks). We found that the absolute difference between $\bar{t}^{Fs}_{rev,Mix}$ and the mean block length in the experiment (20 trials) was significantly smaller than the difference between $\bar{t}^{Fs}_{rev,MB}$ and the experimental mean block length (Mean = 0.935 vs. 8.44, W = 0, p = 1.191e-10). Thus, the Mixture model's inferred reversal times were closer to the task's true block length than those inferred by the MB model. Since the Mixture model also best fit participants' choices, this result suggests that participants were well calibrated to the task structure, reversal times.

We also used the Mixture model as our Backwards model, fitting it to empirical choices and confidences. The Mixture model of confidence fit significantly better than the MF RL characterization of confidence (Mean = .68 vs .745, W = 0,

p = 3.504e-10). The Backward performance based on the Mixture model was highly correlated with QSR (model-agnostic measure of meta-cognitive sensitivity) (r = .69, p = 8.45e-09) (see Fig J in S1 Text). Thus, the better choice model also resulted in a better confidence model and greater consistency with QSR. In addition, the MetaRL.Ratio associated with the Mixture model was independent of performance (r = -.02, p = .886), although still dependent on confidence-bias (r = .49, p = 1.56e-04). Following the same approach as in our model-free analysis of confidence, we incorporated a leaky confidence term [30] into the confidence representation of the Backward model (see Equation 28; Materials and Methods). Although the leaky representation of confidence provided a better fit to empirical choices and confidence ratings compared to the basic Backward model (Mean = .66 vs .68, W = 33.5, p = 1.56e-09), it failed to make the MetaRL.Ratio independent of confidence bias in this case (r = .41, p = .002).

We found that the absolute difference between the Forwards estimate $\bar{t}^{Fs}_{rev,Mix}$ of the mean block length and the actual value in the experiment was significantly smaller than that for the Backwards estimate $\bar{t}^{Bs}_{rev,Mix}$ and the experimental mean block length (Mean = 0.935 vs. 8.286, W = 0, p = 1.191e-10), with the Backward Mixture model expecting later reversals than the Forward Mixture model (see Fig I;J in S2 Text). Thus, the Forward Mixture model provided a closer approximation to participants' experienced block lengths in the experiment than the Backward Mixture model.

## Discussion

We introduced novel measures of metacognitive sensitivity and efficiency for reinforcement learning (RL) problems, and assessed them in data taken from an experiment designed to examine confidence in value-based choices. We assessed sensitivity according to the performance of a Backward model, which synthesized choices that would be consistent with the empirical confidence judgments - and, as such, included a process model for generating confidence [10]. Backward model performance aligned with the model-agnostic metacognitive sensitivity measure, QSR, while offering the advantages of model-based interpretability. Normalizing the performance of the Backward model by the performance of a conventional Forward model fit to the empirical choices lead to a measure of metacognitive efficiency called the MetaRL. Ratio, which, appropriately, did not vary with choice performance. Moreover, our method for scaling the confidence judgments successfully reduced the influence of confidence bias on MetaRL.Ratio, a desirable attribute also sought in other metacognitive efficiency measures such as the M-ratio. Notably, the MetaRL.Ratio remained consistent across different levels of task difficulty, indicating that it captured a stable human characteristic. Our model-based approaches to assessing metacognitive sensitivity and efficiency fill a lacuna in the domain of learning, satisfying crucial criteria for such measures.

To enhance interpretability, we presented our main results based on a simple RL model of choices. However, we also demonstrated that the characteristics of our measures remain consistent when applying more advanced model-free RL and model-based RL (Hidden Semi-Markov) treatments of choice and confidence. In this broader modelling space, a hybrid Mixture model—in which choice probabilities are given by a weighted combination of MF and MB action values—provided the best account of behaviour. Its predictions aligned more closely with the empirical choice and confidence data than those of pure MF RL, and its overall fit exceeded that of MB RL. Moreover, the Mixture model also offered a better description of participants' beliefs about block length (reversal times) than the MB formulation, suggesting that a combination of model-free and model-based computations may better capture human behaviour in this task. Building on this, it would be straightforward to explore other forms of Forward and Backward models (e.g., model-free versus model-based RL treatments as in, [31,32]), including characterizations in which their information is partially different (as in second-order treatments of confidence; [29]). Additionally, confidence representations could be refined using alternative model-derived quantities, such as the Q-value for the chosen option, the absolute difference between Q-values [30,33,34], or the distinction between exploration and exploitation [35].

To allow for differences in the way that participants report confidence, we scaled the model's confidence linearly to align it best with empirical ratings. We anticipated, and indeed observed, that this approach would help diminish the influence of

confidence bias on the performance of the Backward model and the MetaRL.Ratio (along with scaled-QSR). By applying a leaky representation of confidence [29,30], which more closely matched empirical confidence reports, the MetaRL.Ratio became independent of confidence bias within the model-free (MF) reinforcement learning framework - although not in the Mixture model. Thus, the leaky confidence representation did not prove to be a universal solution across all models. It would be interesting to explore whether the dependence of the MetaRL-Ratio on confidence bias could be mitigated using alternative scaling methods or leaky confidence representations. Notably, [16] reported a relationship between metacognitive efficiency and confidence bias, where higher or lower empirical metacognitive efficiency corresponded to higher or lower confidence bias, respectively.

The learning-rate of the Backward model was significantly lower than that of the Forward model, a result we attributed to the auto-correlation in empirical confidence-ratings originally observed by [28,29] and replicated here. Indeed, lower Backward learning-rates were associated with higher auto-correlation (a characteristic that a more sophisticated Backward model could eliminate; see Fig A in S2 Text). Consistently, the learning rate of the Leaky Backward model was higher than that of the Backward model while remaining lower than that of the Forward model (Fig F is S2 Text). In addition, in the low-variance condition, the Leaky Backward learning rate was not associated with the autocorrelation of confidence ratings, whereas in the high-variance condition it remained negatively associated with confidence autocorrelation (Fig A in S2 Text). This indicates that the leaky representation of confidence reduces the dependence of the learning rate on confidence autocorrelation, at least in the low-variance condition. Further work is needed to improve the confidence representation in both task conditions. In simulated data, the parameters of the Backward model were sensitive to artificially-added confidence noise, observed across two different agents (Fig C in S1 Text). The learning-rate decreased at higher levels of noise variance, as expected, because our model of confidence predicted noisier confidence levels less accurately than less noisy ones. Conversely, the inverse temperature decreased under the influence of higher noise variance. Although the interpretation of these results requires further analysis, they were not related to the correlation of learning-rate and inverse temperature at each level of noise (see Fig K in S2 Text).

We observed that the MetaRL.Ratio was not significantly different across levels of task difficulty. In contrast, empirical, Forward and Backward performance and confidence bias were lower in the more difficult, high-variance, condition. [30] assessed the effect of outcome valence on confidence bias, finding higher confidence bias in gain over loss condition (something that we could not assess), even though empirical performance was unaffected. Thus, outcome variance, increased task difficulty, and a negative valence of outcomes decrease metacognitive bias. This raises an important question for future studies regarding the way that task characteristics influence metacognitive bias and efficiency. Our observations, in addition to those of [30,34], provide a more comprehensive understanding of the influence of task conditions on confidence and metacognitive ability. Specifically, we suggest that our measure will prove beneficial in differentiating the impact of task conditions on two aspects of metacognition in the learning domain, extending the anslses of [30,34].

Our parametric RL framework for MetaRL.Ratio enabled us to compare the impact of task difficulty on Forward and Backward parameters, including learning-rate, inverse temperature, and confidence-bound parameters. We observed a slightly lower Forward learning-rate in the high-variance condition compared to the low variance condition, as anticipated due to the influence of aleatoric versus epistemic uncertainty [36,37]. While task difficulty had no impact on the Backward learning rate, it significantly reduced both the Forward and Backward inverse temperatures, with a more pronounced effect on the former and a modest effect on the latter. The reduction in the Forward inverse temperature is consistent with noisier choice; the reduction in Backward inverse temperature would arise if the choice was known by the participants to be noisier, so they reported lower confidence (Fig 6C). However, neither Forward nor Backward model took account of other facets of task difficulty such as the well-known tradeoff between exploration and exploitation [27]. The way that this tradeoff affects confidence reporting is anyhow incompletely clear [35].

It is noteworthy that the impact of task difficulty on Forward parameters was more pronounced than on the Backward model. This finding aligns with the observation that task difficulty affects confidence bias less than empirical performance.

It also demonstrates that the Forward and Backward parameters effectively reflect the influence of task difficulty on two distinct levels of decision-making.

An essential property of a measure of metacognitive efficiency is that it should be consistent across task difficulties, if participants are. We observed that our measures of both metacognitive sensitivity and efficiency were consistent across subjects, something that was not true for empirical or Forward performance. A consistent ordering was also observed in our measure of metacognitive bias. We found that both empirical and modeled representations of confidence bias maintained the perceived order across subjects in both conditions. In a study by [26], a slight correlation of metacognitive sensitivity was reported across different cognitive domains and tasks, while a high correlation was observed for confidence bias, which they referred to as the "finger-print" of participants. Our results, at least in terms of the strengths of correlations, align with those reported by [26]. Despite differences in task difficulties between our study and the diverse domains and tasks in theirs, our findings provide a consistent and broader perspective on the distinction between metacognitive sensitivity and bias. This prompts further investigation into which of these aspects could serve as a more reliable "finger-print" for participants. Subsequent studies could shed light on the neural representation of these two facets of metacognition across various domains and tasks.

Although we concentrated on human metacognition, there is an increasing reliance on automated decision-makers in many domains [38–40]. While these systems can often perform better than humans, they can also err substantially. If they can accurately assess their own performance – i.e., be metacognitively sensitive and efficient – then it would be possible to trust them more appropriately [41,42]. The circumstances under which these systems are used almost inevitably involve varying performance, and so demand a measure such as the one discussed here.

In summary, we introduced a model-based measure of metacognitive efficiency for the learning domain, and validated it on a large dataset of choices on a two-arm bandit task. Our measure was independent of empirical performance, and the way we scaled confidence values reduced the dependency on confidence bias. Our measure of metacognitive sensitivity was also aligned with the QSR. Our measures enabled us to distinguish between two aspects of metacognition across levels of task difficulty, and allowed us to delve into the parametric interpretation of metacognitive behavior.

## Materials and methods

**Ethics statement.** The study was approved by the Social and Societal Ethics Committee (SMEC) of KU Leuven with reference number G-2020–2895-R2(MAR).

**Cognitive task.** The experiment included sixty participants, all of whom were first-year psychology students at KU Leuven participating for course credits. Information about age, gender, and handedness was not recorded and is therefore unknown, although one can typically expect a majority of right-handed, 18-year-old, mostly female participants in this pool. All participants provided informed consent via e-mail and were unaware of the hypotheses of the study.

Data from six participants were removed because their performance remained at chance level in at least one of the blocks, leaving a final sample of fifty-four participants. Participants completed the task on their own PC, with data collected online via the Pavlovia platform (pavlovia.org). Only those using a Windows PC with an external mouse were allowed to take part. They were informed that the goal of the task was to earn as much money as possible on virtual slot machines. On each trial, participants chose between two slot machines—mapped to the S-key (left) and F-key (right)—with one machine offering a higher average reward. They were told that every 18–22 trials, a switch would occur so that the previously worse slot machine would become the better one. After each choice, participants indicated their confidence on a continuous scale ranging from "this was a guess" to "very certain." A numerical reward based on their choice was then displayed, and the next trial commenced. The three highest-scoring participants received a gift voucher from a local store.

In the low-variance condition of the experiment, the rewards following a worse or better choice were drawn from normal distributions $\mathcal{N}(40, 8)$ or $\mathcal{N}(60, 8)$. In the high-variance condition, rewards were drawn from $\mathcal{N}(40, 16)$ or $\mathcal{N}(60, 16)$. These conditions were administered in a counterbalanced order, 12–72 hours apart, and each was followed by a

medium-variance condition that is not analyzed here. Both the low- and high-variance conditions comprised 20 blocks of 18–22 trials, for a total of 400 trials in each condition.

**Exclusion criteria.** To exclude participants performing at or below chance level, we applied a Chi-Square Goodness-of-Fit test separately to the high- and low- variance conditions. This method compares the observed performance (correct vs incorrect responses) to the expected distribution under random guessing (50% correct). The Chi-Square statistic was calculated as:

$$\chi^2 = \sum \frac{(O_i - E_i)^2}{E_i} \tag{1}$$

where $O_i$ is the observed frequency and $E_i$ is the expected frequency under the null hypothesis. We used 1 degree of freedom because the test involves two categories (correct and incorrect responses), with the degrees of freedom calculated as the number of categories minus one. Participants whose performance did not exceed the chi-squared critical value of 3.841 at 1 degree of freedom ($p \geq .05$) — that is, whose choices did not differ significantly from chance (50/50) — were excluded from further analysis. This resulted in the exclusion of six participants.

**Statistical tests.** We employed three types of statistical tests: the non-parametric Wilcoxon test for comparing two groups, the Pearson correlation test to assess the correlation between two variables, and the Z-Fisher test for comparing two correlation results.

## Computational modelling

**Forward model of confidence.** We used maximum likelihood estimation to fit each participant's choices. The model free treatment of choice was Q-learning-based, with learning-rate $\alpha$, and softmax exploration with inverse temperature $\beta$:

$$Q_t(a_t) = Q_{t-1}(a_t) + \alpha(r_t - Q_{t-1}(a_t)) \tag{2}$$

$$P_t(a_t = i; \alpha, \beta) = \frac{\exp(\beta * Q_t(i))}{\exp(\beta * Q_t(1)) + \exp(\beta * Q_t(2))} \quad \text{for } i \in \{1, 2\} \tag{3}$$

which we fit using maximum likelihood:

$$(\alpha^{Fs}, \beta^{Fs}) \sim \underset{\alpha,\beta}{\arg\max} \sum_{t=1}^{N} \log(P_t(a_t^s; \alpha, \beta)) \tag{4}$$

where $a_t^s$ are the empirical choices of participant $s$. We evaluated the performance of the best-fitting Forward model by the frequency with which it chose the option with higher average reward, $\overline{\text{Perf}}^{Fs}$ for participant $s$, when run autonomously on the task using the same (potential) reward sequences as the participants experienced in the experiment. This is as we would implement for a posterior predictive check of the fit of a model.

To model Forward confidence, we scaled the output of the softmax (written as $P_t^{Fs}(a_t^s)$ for participant $s$) using parameters $L$ and $H$, both in the range [1,5] and subject to $L < H$:

$$\text{Conf}_t^{Fs}(L, H) = (H - L) * P_t^{Fs}(a_t^s) + L \tag{5}$$

which we fit by minimizing the Euclidean distance between the scaled and empirical ($\text{Conf}_t^s$) confidence (using scipy.optimize.minimize).

$$(L^{Fs}, H^{Fs}) \sim \arg \min_{L,H} \sum_{t=1}^{N} \left( \text{Conf}_t^{Fs}(L, H) - \text{Conf}_t^{s} \right)^2$$

(6)

In the above equation, $\text{Conf}_t^{Fs}(L, H)$ and $\text{Conf}_t^{s}$ are both in the range [1,5].

**Backward model of confidence.** For the case of perceptual decision-making, the metacognitive sensitivity measure meta-$d'$ comes from treating the empirical reports of confidence as the result of a probabilistic choice process (as in a first-order decision-making model), and quantifying the effective perceptual sensitivity of that model. This can be seen as going backwards from confidence to choice. We therefore defined a Backward model (Fig 1, dark-red) in which we characterize participants' choices as coming from the same RL process as the Forward model, but with decision-making parameters fit to make the (similarly scaled) choice probabilities match the empirical confidence judgments as best as possible (rather to match the empirical choices) (Equations 9; 10). We then evaluated the performance $\overline{\text{Perf}}^{Bs}$ of the best-fitting Backward model for participant $s$ when run autonomously on the task, as for the Forward model. In full:

$$Q_t(a_t) = Q_{t-1}(a_t) + \alpha(r_t - Q_{t-1}(a_t))$$

(7)

$$P_t(a_t = i; \alpha, \beta) = \frac{\exp(\beta * Q_t(i))}{\exp(\beta * Q_t(1)) + \exp(\beta * Q_t(2))} \quad \text{for } i \in \{1, 2\}$$

(8)

and, writing this as $P_t^{Bs}(a_t^s)$ for the backwards model for participant $s$, the modeled confidence is:

$$\text{Conf}_t^{Bs}(L, H) = (H - L) * P_t^{Bs}(a_t^s) + L$$

(9)

in which $L$ and $H$, $L < H$, were parameters with the same restriction as the equivalent parameters in the Forward model. The parameters were fit by minimzing the Euclidean distance between the confidence values predicted by the model and those reported by the subjects:

$$(\alpha^{Bs}, \beta^{Bs}, L^{Bs}, H^{Bs}) \sim \arg \min_{\alpha,\beta,L,H} \sum_{t=1}^{N} \left( \text{Conf}_t^{Bs}(L, H) - \text{Conf}_t^{s} \right)^2$$

(10)

noting the dependence of $P_t^{Bs}(a_t^s)$, and thus $\text{Conf}_t^{Bs}$ on $\alpha, \beta$.

**Measure of metacognitive efficiency.** Inspired by the M-ratio for perceptual-decision-making, we defined the MetaRL. Ratio = $\overline{\text{Perf}}^{Bs}/\overline{\text{Perf}}^{Fs}$ as a measure of metacognitive efficiency.

**Fitting and simulations.** The fitting for all the analyses described in the "Results" section or in "S1 Text" and "S2 text" was repeated 120 times to stabilize the estimates. Using the fitted parameters for each subject, we then averaged over 100 simulations to estimate model performance. This averaging accounts for the randomness inherent in the models due to the softmax choice function. All analyses were implemented in Python using the `scipy.optimize.minimize` module to minimize the negative log-likelihood for each subject. We employed consistent parameter ranges across subjects and models (both forward and backward), with $\alpha \in [0, 1]$, $\beta \in [0, 100]$, $L \in [1, 5]$, and $H \in [1, 5]$ under the constraint $L < H$.

**Scaled-QSR.** Given empirical choices $a_t^s$ (when the better bandit is $b_t^s$), the trial by trial performance was 1 if the empirical choice was the better bandit and 0 otherwise.

$$I_t^s = \begin{cases} 1, & \text{if } a_t^s = b_t^s, \\ 0, & \text{if } a_t^s \neq b_t^s. \end{cases}$$

(11)

The confidence values were $\text{Conf}_t^s$ for participant $s$ and the (normalized) QSR was

$$\text{QSR}^s = 1 - \frac{1}{N} \sum_{t=1}^{N} \left( \frac{\text{Conf}_t^s - 1}{4} - I_t^s \right)^2 \tag{12}$$

In the above equation, confidence in the range [1,5] is normalized to be comparable with performance. Without normalization, if confidence ratings had a different scale (e.g., [1,6] or [0,100]), the QSR could overweight or underweight one metric relative to the other. Then, we scaled the empirical confidence values according to the same procedure as for the Forward and Backward models (Equations 5 and 9):

$$\text{Conf}_t^{Qs}(L, H) = (H - L)\,\text{Conf}_t^s + L \tag{13}$$

and determined confidence bounds by finding

$$(L^{Qs}, H^{Qs}) = \arg\max_{L,H} \left\{ 1 - \frac{1}{N} \sum_{t=1}^{N} \left( \frac{\text{Conf}_t^{Qs}(L, H) - 1}{4} - I_t^s \right)^2 \right\}. \tag{14}$$

We call the maximum value of the expression on the right of Equation 14 the scaled-QSR.

**Alternative model of choice.** We also explored our metacognitive measures using a more powerful RL algorithm as both Forward and Backward models. This separates learning rates for the chosen action $a_t$ on trial $t$ according to whether the prediction error

$$\delta_t = r_t - Q_{t-1}(a_t) \tag{15}$$

is positive or negative:

$$Q_t(a_t) = Q_{t-1}(a_t) + \begin{cases} \alpha^+ \delta_t & \text{if } \delta_t > 0 \\ \alpha^- \delta_t & \text{if } \delta_t < 0 \end{cases} \tag{16}$$

and also uses counterfactual learning, adjusting the $Q$-value for the unchosen action $\tilde{a}_t = 3 - a_t$ with its own (typically negative) learning rate $\alpha^u$:

$$Q_t(\tilde{a}_t) = Q_{t-1}(\tilde{a}_t) + \alpha^u \delta_t \tag{17}$$

on the grounds that the participants know that the bandits are anticorrelated - so if they find one better than expected, then the other one should be worse than expected. This four-parameter model fits more proficiently than the simpler, two-parameter Q-learning model (see Fig D in S1 Text).

**Model-Based decision-making; Hidden Semi Markov Model (HSMM).** In the Model-Based (MB) framework the latent state of the environment is inferred using an HSMM that explicitly accounts for state durations. Here, the two latent states (indexed by $S \in \{1, 2\}$) represent which action has the higher reward. In state 1, action 1 has higher reward, and in state 2, action 2 has higher average reward.

We denote the sequence of latent states as $S_{1:t}$ from time 1 to $t$. Let random variable $h$ be the time already spent in the current state, and random variable $d$ be the total duration of the current state. Then the joint probability of the latent states, the past rewards, and the duration can be expressed as follows for $i \neq j$:

$$P(S_{t-h+1:t} = i, r_{1:t-1}, h, d \mid a_{1:t-1}, \theta) =$$
$$P(r_{t-h+1:t-1} \mid S_{t-h+1:t-1} = i, a_{t-h+1:t-1}, \theta)$$
$$P(S_{t-h} = j, r_{1:t-h} \mid a_{1:t-h}, \theta) P(h \mid d) P(d \mid \theta), \tag{18}$$

where $\theta = (D, n, p, \mu_S^a, \sigma_C)$ denotes the full set of model parameters: $D$ is the maximum possible duration of a latent state; $n$ and $p$ are the parameters of the negative binomial duration distribution (Eq. 22). is the mean reward associated with state $S$ and action $a$; and $\sigma_C$ is the standard deviation of the Gaussian reward distribution, which depends on the task condition (high-variance or low-variance). By marginalizing out $d$ and $h$ random variables, we obtain:

$$P(S_t = i, r_{1:t-1} \mid a_{1:t-1}, \theta) =$$
$$\sum_{d=1}^{D} P(d \mid \theta) \sum_{h=1}^{d} P(h \mid d) P(S_{t-h} = j, r_{1:t-h} \mid a_{1:t-h}, \theta)$$
$$P(r_{t-h+1:t-1} \mid S_{t-h+1:t-1} = i, a_{t-h+1:t-1}, \theta). \tag{19}$$

We define the *forward variable*, $\alpha(t, j)$, as the joint probability of the observed rewards from time 1 to $t$, denoted as $r_{1:t}$, and the event that state $j$ ended at time $t$ conditioned on $a_{1:t}$ and $\theta$. It can be computed recursively for any time $t$ and state $j$:

$$\alpha(t, j) := P(S_t = j, S_{t+1} = i, r_{1:t} \mid a_{1:t}, \theta)$$
$$= \sum_{d=1}^{D} P(d \mid \theta) \alpha(t - d, i) P(r_{t-d+1:t} \mid S_{t-d+1:t} = j, a_{t-d+1:t}, \theta). \tag{20}$$

Using Eq. 20 and assuming a uniform prior $P(h \mid d) = \frac{1}{d} . h \leq d$ (and 0 for $h > d$), Eq. 19 simplifies to:

$$P(S_t = i, r_{1:t-1} \mid a_{1:t-1}, \theta) =$$
$$\sum_{d=1}^{D} P(d \mid \theta) \sum_{h=1}^{d} \alpha(t - h, j) P(r_{t-h+1:t-1} \mid S_{t-h+1:t-1} = i, a_{t-h+1:t-1}, \theta), \tag{21}$$

**Duration Modeling.** We assume that a latent state persists for a duration $d$ (with $d \in \{1, \ldots, D\}$, and in our implementation $D = 70$, as the maximum number of consecutive identical actions observed across subjects and conditions was 68). The duration $d$ is modeled using a truncated negative binomial distribution:

$$P(d \mid D, \theta) = \frac{\binom{d+n-1}{d} p^n (1-p)^d}{\sum_{d=1}^{D} \binom{d+n-1}{d} p^n (1-p)^d}. \tag{22}$$

The negative binomial distribution, $NB(n, p)$, can be interpreted as the distribution over the number of heads observed before a fixed number of tails occur, under repeated Bernoulli trials with tail probability $p$. Thus, the duration $d$ corresponds to the number of heads observed before $n$ tails are generated. The geometric distribution is a special case of the negative binomial distribution with $n = 1$. The mean of truncated $NB(n, p)$ represented the model estimation of reversal trials, $\bar{t}_{rev}$. For each subject, the parameters $n$ and $p$ were fitted using a linear grid search: $p$ was varied linearly from 0.01 to 0.99 in steps of approximately 0.06, and $n$ from 1 to 100 in steps of 6.

**Emission Model.** This is the probability of a sequence of observed rewards given the latent states and actions in each trial ($P(r_{t:t+d} \mid S_{t:t+d}, a_{t:t+d}, \theta)$). Since the rewards in different trials are conditionally independent given the states and actions, we can factorize the joint probability as:

$$P(r_{t:t+d} \mid S_{t:t+d},\, a_{t:t+d},\, \theta) = \prod_{k=t}^{t+d} P(r_k \mid S_k, a_k, \theta),$$

(23)

where $P(r_k \mid S_k, a_k, \theta)$ is modeled as a Gaussian distribution with the same mean ($\mu_S^a$) and variance ($\sigma_C^2$) as in the task, means of 40 and 60, and variances of 8 or 16. Without loss of generality, we assume that in state $S_t = 1$, action $a_t = 1$ yields a higher average reward, and the reverse holds for $S_t = 2$. Then we have:

$$P(r_t \mid s_t, a_t, \theta) = \begin{cases} \mathcal{N}(r_t; \mu_S^a = 60, \sigma_C^2) & \text{if } S_t = a_t, \\ \mathcal{N}(r_t; \mu_S^a = 40, \sigma_C^2) & \text{if } S_t \neq a_t, \end{cases}$$

(24)

where $\sigma_C^2$ depends on whether the task condition is high or low variance.

Using Eq. 21, we can compute the posterior probability of each state:

$$P(S_t \mid r_{1:t-1},\, a_{1:t-1},\, \theta) = \frac{P(S_t, r_{1:t-1} \mid a_{1:t-1},\, \theta)}{\sum_{S_t'} P(S_t', r_{1:t-1} \mid a_{1:t-1},\, \theta)}$$

(25)

Finally, we define the probability of each action corresponding to the probability of the state in which that action has a higher average reward. For example, if we set in state 1, the $a_t = 1$ has a higher reward, and in state 2, the $a_t = 2$ has a higher reward, then we have:

$$P_{MB}(a_t = 1 \mid S_t) = P(S_t = 1 \mid r_{1:t-1}, a_{1:t-1}, \theta) \quad P_{MB}(a_t = 2 \mid S_t) = P(S_t = 2 \mid r_{1:t-1}, a_{1:t-1}, \theta)$$

(26)

**Mixture Model, combination of both MF and MB models.** The MB probabilities $P_{MB}(a|S)$ (obtained as shown above) are combined with the model-free (MF) probabilities computed via a softmax function over $Q$-values (Equation 8). In our simulation the overall choice probability is a weighted mixture:

$$P_{Mix}(a_t \mid S_t) = \gamma\, P_{MF}(a_t) + (1 - \gamma)\, P_{MB}(a_t \mid S_t),$$

(27)

where $\gamma$ is the mixture weight.

Thus the MB probabilities are mixed with MF probabilities to simulate an action response, and the HSMM forward variable is updated with the new reward information for subsequent trials. The fitting method for the mixture model was the same as the one for the HSMM.

**Leaky model of confidence.** A leaky term was added to representation of confidence in the Backward model [29,30], based on both MF and Mixture choice models, thus the term $\text{Conf}_t^{Bs}(L, H)$ (9) was combined linearly with $\text{Conf}_{t-1}^{Bs}(L, H)$:

$$\text{ConfL}_t^{Bs} = \lambda\, \text{Conf}_t^{Bs} + (1 - \lambda)\, \text{Conf}_{t-1}^{Bs}$$

(28)

represents a leaky integration of confidence across trials.

Here, $\text{Conf}_t^{Bs}$ denotes the current trial's confidence (after applying a linear transformation to the response probability), $\text{Conf}_{t-1}^{Bs}$ represents the previous trial's confidence, and $\lambda$ is a parameter between 0 and 1 that determines the weighting between the current and previous confidence.

This equation means that the confidence estimate for the current trial, $\text{ConfL}_t^{Bs}$, is not solely based on the current response probability but is also influenced by the confidence from the previous trial. When $\lambda = 1$, only the current confidence is considered, and when $\lambda = 0$, the system fully relies on the previous confidence value. For intermediate values, the estimate is a weighted average, creating a smooth, leaky integration over time.

## Supporting information

**S1 Text. Supplementary information.** This supplementary text contains Boxes *A* and *B* and Figures *A − K*. **Fig A. Recovery analysis for Forward and Backward models.** A) In the Forward model, we simulated the behavior of 100 agents using random values for four parameters: learning-rate ($\alpha$), inverse temperature ($\beta$), lower-bound for confidence (*L*), and upper-bound for confidence (*H*). We also randomly selected the trajectory of rewards from those experienced by the subjects. Subsequently, we fit the behavior of these agents to the Forward model to recover the parameters. The original values of each of the four parameters were highly correlated with their corresponding recovered values across all agents. B) The same analysis was conducted for the Backward model, based on the confidence ratings of the agents in A. Again, actual and recovered parameters were significantly correlated. The dots in the plots above indicate the respective estimations for each subject. **Fig B. The influence of different fitting methods on recovered parameters** The 4 recovered parameters; learning-rate (A), inverse-temperature (B), confidence upper (C) and lower bounds (D), were highly correlated between Forward and Backward models. The above plots display dots that represent the relevant estimations for each subject. **Fig C. Forward (blue) and Backward (red) performance** under influence of confidence noise. A) We created synthetic agents using empirical choices but synthetic confidence ratings. In these agent, called Hmeta($\sigma^2$), we first assigned the highest confidence to correct choices and the lowest to incorrect ones, and then corrupted these values by Gaussian noise with variance $\sigma^2$ (suitably truncated and quantized). We observed that increasing the level of confidence noise in Hmeta($\sigma^2$)'s confidence values led to a decrease in Backward performance and constant Forward performance, as was expected. B) The Backward learning-rate decreased with increasing noise variance. C) The inverse-temperature increased as noise variance increased. D-F) We repeated the same procedure for simulated choices and confidence ratings coming from the Forward model fit to the same empirical data. Similar to our findings with the Hmeta($\sigma^2$) agent, we observed a decrease in Backward performance under the influence of confidence noise, as expected, while Forward performance remained constant. E) The Backward learning-rate for the Forward($\sigma^2$) agent decreased with higher noise variance. F) The inverse-temperature increased as noise variance increased. The dots in the above plots denote the related estimations for each subject. **Fig D. Comparison between simpler and more complex Forward model fits.** The simple, two-parameter Forward model (1), used in most of the text, fits the behavioral data less well than the more complicated four-parameter Forward model (2) was a better model of choice relative to the simple model. The dots in the plots above indicate the respective estimations for each subject. **Fig E. Metacognitive measures for more complex Forward model.** A) The QSR and Backward performance were positively correlated (r = 0.35, p = .008). B) The confidence scaling method decreased the dependency of MetaRL.Ratio on confidence-bias (Z = -3.485, p = 0.0004). C, D, E, F & G) The MetaRL.Ratio was independent of empirical performance (r = 0.01, p = .943), and all Forward parameters (Positive learning-rate; r = 0.00, p = .993, Negative learning-rate; r = -0.01, p = .938, unchosen learning-rate; r = 0.26, p = .057, and $\beta$; r = 0.00, p = .997). The dots in the above plots denote the related estimations for each subject. **Fig F. MetaRL.Ratio and confidence-bias across levels of difficulty for the more complex Forward model.** A) the MetaRL.Ratio was not different between difficulties (W = 641.0, p = .382); B) and nor was the average confidence (W = 540.0, p = .081). **Fig G. Comparison between MF, Mixture, and MB models in low-variance condition of task.** A) The goodness of fit, as measured by BIC, was not significantly different between MF and the Mixture model. However, both the Mixture model and MF fit better than the MB model. B) According to AIC, the Mixture model provided a better fit than both MF and MB, while MF fit our empirical data better than MB. **Fig H. Comparison between empirical performance and models performance in the low-variance condition of the task.** The performance of MB RL and Mixture models was significantly closer to participants' performance than MF RL (MB RL vs. MF RL: 0.87 vs. 0.76, W = 0, p = 1.626e-10; Mixture vs. MF RL: 0.85 vs. 0.76, W = 0, p = 1.626e-10). However, while the Mixture model's performance was significantly different from empirical performance (Mean = 0.85 vs. 0.86, W = 312, p = 2.1e-04), MB performance was not significantly different from empirical performance (Mean = 0.87 vs. 0.86, W = 686, p = 9.818e-01). **Fig I. Comparison between Forward and Backward Mixture models in choice, confidence and parameters in**

**low-variance condition.** A) The performance of the Backward Mixture model was significantly lower than both empirical and Forward performance. Additionally, Forward performance significantly lagged behind empirical performance. B) The confidence-bias of the Backward model, quantified as empirical average confidence, was not significantly different from that of the Forward model or empirical data, while there was a significant difference between the confidence-bias of the Forward model and empirical data. C) The Forward model predicted choices better than the Backward model, as measured by the negative log likelihood. D) The confidence ratings of the Backward model were closer to the empirical data than those of the Forward model. E) The learning-rate was significantly lower in the Backward model compared to the Forward model. F) The inverse-temperature was significantly higher in the Backward model compared to the Forward model. The dots in the plots above represent the corresponding estimations for each subject in the low variance condition of task. **Fig J. Relationship with Quadratic Scoring rule, model-agnostic measure of meta-cognitive sensitivity in low-variance condition of task.** A) Backward performance based on the Mixture model was not significantly correlated with QSR. B) Backward performance was also not significantly correlated with scaled-QSR, which determines a linear scaling of empirical confidence values to maximize QSR. The dots in the plots above represent the corresponding estimations for each subject. Each dot in the above plots reflects the associated estimations for the subjects in the low variance condition of task. **Fig K. Independence of the MetaRL.Ratio, based on Mixture model, from empirical choice parameters in low-variance condition of task.** A) The MetaRL.Ratio, our measure of metacognitive efficiency, was independent of empirical performance. B) The correlation between MetaRL.Ratio and confidence-bias decreased after applying the confidence scaling method. C & D) The MetaRL.Ratio was not significantly correlated with the inverse-temperature (C) or the learning-rate (D) of the Forward Mixture model. The above plots display dots that represent the relevant estimations for each subject in the low variance condition of task.
(PDF)

**S2 Text. Supplementary information.** This supplementary text contains Figures $A - T$. **Fig A. Backward learning-rate and auto-correlation in confidence-rates.** A&B) Task conditions are shown in order (low-variance, then high-variance). In the Backward model, learning rate was negatively correlated with the autocorrelation of confidence ratings, indicating that lower learning rates were associated with more autocorrelated confidence ratings: low-variance ($r = -0.27, p = 4.93e - 02$) and high-variance ($r = -0.46, p = 4.68e - 04$). C&D) The same analysis for the Leaky Backward model. In the low-variance condition, the correlation was not significant ($r = -0.17, p = 2.28e - 01$), whereas in the high-variance condition, lower learning rates were associated with higher autocorrelation ($r = -0.40, p = 2.50e - 03$). Each dot represents one participant. The strength of the correlation did not differ significantly between the Backward and Leaky Backward models in either condition (low-variance: $Z = 0.53, p = .59$, high-variance: $Z = 0.37, p = 0.71$). **Fig B. Forward and Backward confidence bound parameters.** Each dot in the above plots reflects the associated estimations for the subjects. **Fig C. The scaled QSR was less dependent on confidence-bias relative to QSR.** The dots shown in the above plots correspond to the estimations for each subject. **Fig D. Independence of MetaRL.Ratio from Forward parameters in synthetic data.** A) The learning-rate was fixed at three levels, the MetaRL.Ratio remained around 1 for 10 equalized distant points of $\beta$ between 5 and 90. B) The $\beta$ was fixed at three levels, the MetaRL.Ratio remained around 1 for 10 equalized distant points of $\beta$ between 0.05 and 1. **Fig E. Leaky model of confidence; independence from confidence bias.** We also considered an improved, 'Leaky' model of confidence which captures autocorrelation in confidence reports by using a weighted average of the 'true' confidence on the current trial and the confidence report on the previous trial. A) For some subjects the MetaRL.Ratio from leaky and non-leaky model of confidence were different from each other. B) When using this leaky model, the MetaRL. Ratio was no longer significantly correlated with the confidence bias. C) The weight accorded to the confidence report on the previous trial, called the 'Last Confidence Multiplier', was positively correlated with the degree of autocorrelation in the confidence reports. **Fig F. Comparison of learning rates between Forward, Backward, and Leaky Backward models.** The Backward model had a lower learning rate than the Forward model, and the Leaky Backward model had a higher learning rate

than the Backward model while remaining lower than the Forward model. **Fig G. Confidence bound parameters were not influenced by task difficulty.** A & B) The lower and upper bounds of confidence ($L$ and $H$) and C) their average were not influenced by task difficulty. The dots in the above plots denote the related estimations for each subject. **Fig H. Parameters of the Forward MB model** A & B) Fitted $n$ and $p$ parameters of the Negative Binomial distribution for the Forward MB model. C) The corresponding truncated Negative Binomial distribution (capped at 70 trials). The vertical blue dashed line indicates the mean of this distribution, representing the model's reversal trials. In contrast, the vertical black dashed line indicates the mean reversal trial across all subjects and blocks in the empirical data. **Fig I. Parameters of the Forward Mixture model.** A & B) Fitted $n$ and $p$ parameters of the Negative Binomial distribution for the Forward Mixture model. C) The corresponding truncated Negative Binomial distribution (capped at 70 trials). The vertical blue dashed line indicates the mean of this distribution, representing the model's reversal trials. In contrast, the vertical black dashed line indicates the mean reversal trial across all subjects and blocks in the empirical data. **Fig J. Parameters of the Backward Mixture model.** A & B) Fitted $n$ and $p$ parameters of the Negative Binomial distribution for the Backward Mixture model. C) The corresponding truncated Negative Binomial distribution (capped at 70 trials). The vertical blue dashed line indicates the mean of this distribution, representing the model's reversal trials. In contrast, the vertical black dashed line indicates the mean reversal trial across all subjects and blocks in the empirical data. **Fig K. Correlation between parameters.** A) In the Forward model, the learning rate and inverse temperature were not correlated ($r = -0.16, p = 0.235$). B) However, a negative correlation was observed for these parameters in the Backward model ($r = -0.61, p < 0.001$). C) Specifically, in the H-meta agent, these two Backward parameters exhibited a negative correlation for the noiseless model ($r = -0.40, p = .003$). D) This negative correlation was not evident at a standard deviation of 4 ($r = -0.26, p = .058$). E) For the Forward agent, the correlation in confidence noise with a standard deviation of 0 was not significant ($r = -0.16, p = .237$). F) In contrast, a significant negative correlation was found for confidence noise with a standard deviation of 4 ($r = -0.58, p < 0.001$). The dots in the above plots denote the related estimations for each subject. **Fig L. Number of required iterations for fitting stability.** Mean of (A) Negative Log-Likelihood (LL) and (B) Confidence Distance Error are shown in this plot. The fitting process involves a non-convex optimization problem, where both choice and confidence are fitted to the subject's responses over several iterations, selecting the best-fitting parameters. This plot illustrates the mean absolute differences between the best-fitting parameters at iteration t and the results after 200 iterations. It is evident that the fitting stabilizes for both measures after approximately 120 iterations, which was chosen as the cutoff threshold for generating other plots. The y-axis is on a log scale, and the error bars represent 95% confidence intervals. **Fig M. Comparison between three representation of confidence.** A) The Backward model, incorporating the absolute difference between Q-values (Model3), exhibited a significantly superior fit to the empirical data compared to the Backward model that represented confidence using the Probability of choice (Model1). Interestingly, no significant difference was observed between Probability of choice and the Q-value of the chosen option (Model2). Each dot reflects the associated estimations for the subjects. B) The Probability of choice resulted in the most sensitive Backward performance to confidence noise compared with the other representations we tested. **Fig N. Comparison between Forward and Backward models in choice, confidence and parameters in high-variance condition.** A) The performance of the Backward model was significantly lower than both empirical and Forward performance. Additionally, Forward performance significantly lagged behind empirical performance. B) The confidence-bias, quantified as empirical average confidence, levels of the Backward model were not significantly different from the Forward model and empirical data, while there was a significant difference between the confidence-bias of the Forward model and empirical data. C) The Forward model predicted choices better than the Backward model, as measured by the negative log likelihood. D) The confidence ratings of the Backward model were closer to the empirical data than those of the Forward model. E) The learning-rate was significantly lower in the Backward model compared to the Forward model. F) The inverse-temperature parameter was not significantly different between two models. The dots in the plots above represent the corresponding estimations for each subject in the high

variance condition of task. **Fig O. Relationship with Quadratic Scoring rule, model-free measure of meta-cognitive sensitivity in high-variance condition of task.** A) Backward performance was not significantly correlated with QSR. B) Backward performance was also not significantly correlated with scaled-QSR, which determines a linear scaling of empirical confidence values to maximize QSR. The dots in the plots above represent the corresponding estimations for each subject. Each dot in the above plots reflects the associated estimations for the subjects in the high variance condition of task. **Fig P. Relationship between MetaRL.Ratio and empirical choice parameters in high-variance condition of task.** A) The MetaRL.Ratio, our measure of metacognitive efficiency, was independent of empirical performance. B) The correlation between MetaRL.Ratio and confidence-bias decreased after applying the confidence scaling method. C & D) The MetaRL.Ratio was not significantly correlated with the inverse-temperature (C) or the learning-rate (D) of the Forward model. The above plots display dots that represent the relevant estimations for each subject in the high variance condition of task. **Fig Q. Relationship between Backward performance and confidence bias in low- and high-variance conditions.** Backward performance was not independent of confidence bias in the low-variance condition, whereas it was independent in the high-variance condition. **Fig R. Correlation of empirical performance and MetaRL.Ratio between low- and high-Variance Conditions.** Empirical and Forward model performance were not correlated between the two task conditions. In contrast, both the Backward performance (red) and MetaRL.Ratio (purple) were. **Fig S. Correlation of confidence-bias between Low- and High-Variance Conditions.** The confidence-bias was correlated between two task difficulties for empirical data and also Forward and Backward models. **Fig T. The required number of trials for stability of our measure.** The difference of MetaRL.Ratio across all trials, called MetaRL.Ratio$_T$, from MetaRL.Ratio$_t$ (the estimate of the MetaRL.Ratio from trial 10 to t (10 < t)) was utilized as an estimation of stability of our measure. The MetaRL.Ratio$_t$ got closer to MetaRL.Ratio$_t$ after trial 100 and the variance decreased considerably after trial 280, while there were 400 trials in the task.
(PDF)

## Acknowledgments

Thanks to Monica De Bock for help with data collection.

## Author contributions

**Conceptualization:** Sara Ershadmanesh, Peter Dayan.

**Data curation:** Kobe Desender.

**Formal analysis:** Sara Ershadmanesh, Ali Gholamzadeh.

**Funding acquisition:** Peter Dayan.

**Investigation:** Sara Ershadmanesh, Ali Gholamzadeh, Peter Dayan.

**Methodology:** Sara Ershadmanesh, Ali Gholamzadeh, Peter Dayan.

**Project administration:** Sara Ershadmanesh, Peter Dayan.

**Resources:** Peter Dayan.

**Software:** Sara Ershadmanesh, Ali Gholamzadeh, Peter Dayan.

**Supervision:** Sara Ershadmanesh, Peter Dayan.

**Validation:** Sara Ershadmanesh, Ali Gholamzadeh, Kobe Desender, Peter Dayan.

**Visualization:** Sara Ershadmanesh, Ali Gholamzadeh, Peter Dayan.

**Writing – original draft:** Sara Ershadmanesh, Ali Gholamzadeh, Peter Dayan.

**Writing – review & editing:** Sara Ershadmanesh, Ali Gholamzadeh, Kobe Desender, Peter Dayan.

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
