## [Decision Letter · Decision Letter 0]

24 Jul 2024

Dear Dr. Ershadmanesh,

Thank you very much for submitting your manuscript "Metacognitive Efficiency in Learned Value-based Choice" for consideration at PLOS Computational Biology.

As with all papers reviewed by the journal, your manuscript was reviewed by members of the editorial board and by several independent reviewers. In light of the reviews (below this email), we would like to invite the resubmission of a significantly-revised version that takes into account the reviewers' comments.

The Reviewers have noted a number of concerns in varying degrees, ranging from minor to significant. I encourage you to consider their comments thoroughly in your revision and look forward to receiving the revised manuscript.

We cannot make any decision about publication until we have seen the revised manuscript and your response to the reviewers' comments. Your revised manuscript is also likely to be sent to reviewers for further evaluation.

Sincerely,

Megan A. K. Peters, Ph.D.

Academic Editor

PLOS Computational Biology

Daniele Marinazzo

Section Editor

PLOS Computational Biology

The Reviewers have noted a number of concerns in varying degrees, ranging from minor to significant. I encourage you to consider their comments thoroughly in your revision and look forward to receiving the revised manuscript.

Reviewer's Responses to Questions

**Comments to the Authors:**

Reviewer #1: Review of "Metacognitive efficiency in learned value-based choice"

Reviewer: Michael Landy

This paper develops an analogue of M-ratio for a bandit choice task that the authors call MetaRL.Ratio. The analogy is needed because in this value-based task with changing conditions (the average payoffs swap between bandits every 20 trials or so) the participant's estimates of value are constantly changing so that, effectively, the d' for which machine is preferable is also constantly changing. The paper examines several competing models as well, and shows that MetaRL.Ratio has much of what would desire in a statistic that measures metacognitive efficiency independently of first-order sensitivity. I've worked some in metacognition in recent years, but not in RL, so I'll take the authors' word that work on metacognition for RL tasks is new. It's an excellent first effort in that direction: clear and thorough. My comments are mainly about clarity.

Specifics (page/para/line):

2/3/16: This is the first mention of QSR and of the Brier score. Since neither is defined here, mentioning that QSR is 1 minus the Brier score is kind of useless for the reader who is unfamiliar with either (which I was as I read it). Now that I know what they are, the description of the QSR as "model free" seems incorrect in the current context. The Brier score is a difference of estimated and actual probability (per trial here) and it seems to me that the "estimate" here can only come from a model of the participant's behavior, so not exactly model-free.

4/1/4: The switches happen every 18-22 trials and you tell the participant that is the situation. Thus, an ideal-performance model would include counting trials and anticipating a change only starting with the 18th trial (and averaging the values up until then). After the first switch, the time of switch is uncertain, but similar ideal principles apply (somewhat analogous to my paper on categorization with randomly switching priors, with Weiji Ma, Elyse Norton and Luigi Acerbi). You mainly consider two variants of Q-learning as well as the QSR, all of which ignore this knowledge of the participants. I was a little surprised by the lack of discussion of possible models that take this knowledge into account.

5/1/2: "worse choice Backward performance" doesn't parse. Maybe delete "choice"?

5/4/4: I suppose you have to say "slightly higher" given a significant p-value (seems suspect, but so it goes...), but really a 1/4% difference is not significant (in the common English meaning of the word). Oy!

The supplementary figures are not in the order in which they are cited in the main text. Also Supp. 3F is cited before 3E.

6/4/5: you saw "The Scaled-QSR was not correlated...", but then cite a statistic that indicates that it was correlated. Delete "not"?

Figure 4 and other scatterplots: Are the dots individual subjects? Pairings of subjects and variance condition? Legends should clarify this throughout.

8/1/4: I would vote for dropping the scientific notation for p-values when the are larger than .001 (as in "p=.06").

8/2/8: hovered around a value of 1 AS WE VARIED Learning-rate...

9/2/1: was higher in THE high- versus THE low-variance condition

9/last sentence: Much of this page feels like a data dump rather than a description of results that impact a previously stated hypothesis. This last sentence is, to me, a non-conclusion conclusion (it doesn't say what the results mean concerning value-based meta-cognition, but only that you were able to measure stuff).

Figure 6: The legend describes green vs. blue backwards from the axis labels in the plot, and the key is really obscure for figuring out which violin plots correspond to low vs. high variance. I suggest you put the words "Low" and "High" below the corresponding columns and maybe "Variance:" to the left.

Tables 1 and 2: The values in the tables all (mostly?) appear in the running text, making the tables (almost) redundant. I don't understand why these correlations didn't deserve a scatterplot (possibly in the Supplement if you prefer) but all the other correlations did.

12/2: Is the Supplement figure corresponding to this paragraph ever cited in the main text? I didn't check to make sure that all Supplementary figures were cited.

14: There were a lot of experimental details omitted from the Methods section: description of the participants (how many, age, etc.), informed consent, what they were paid and whether payment was increased in some way by single-trial or overall RL performance and details of the trial-by-trial feedback.

14/5/5: that we do NOT analyze

15/1: I can't guess what this chi-squared test is nor what criterion you use on its results to merit exclusion.

15/3/after Eq. 3: "when run autonomously on the task using the same (potential) reward sequences as the participants": I found this unclear. The RL algorithm is deterministic, so I'm guessing that you re-run the model on sequences of pairs of payoffs that are newly sampled from the appropriate Gaussian distributions and do that a bunch of times to get average performance, etc. BUT, the quoted sentence is unclear to me whether it means you ran the model ("?autonomously") only on the sequence the participant experienced or on new ones.

Eqs. 8-9: First, I found the jump from H/L in Eq. 8 to H^{B,s}/L^{B,s} a bit jarring. H/L sounds like a fixed pair of numbers (the highest and lowest confidence the subjects used), whereas H^{B,s}/L^{B,s} sound like free parameters (possibly constrained to be at least as wide as the participant's range?). Also, you call the cost function "maximum likelihood", which implies you are thinking of additional Gaussian noise on produced confidence settings. Gaussian noise is a bit sketchy, since the range is constrained.

16/3 (Fittings and simulations): The explanation for 120 fits and 100 simulations is unclear. Are 120 fits done to try to find a global minimum (and not get stuck in local minima)? Is this non-convex? Are the 100 simulations because you are, in fact, re-drawing the random payoffs on every simulation? Please clarify.

Eq. 10: For me, the \chi_{a=b} notation is non-standard. I might have used Kronecker \delta instead, but would have explained what it meant. Please clarify.

Eqs. 11-12: Something seems wrong here. Conf_t^s is the empirical response (ranging from 1 to 4), so the normalization in Eq. 10 scales it to look range from 0 to 1. It makes no sense to then scale it in Eq. 11 and in Eq. 4. It sounds like you want, in Eq. 11, Conf_t^s to be a number that ranges from zero to one, i.e., you want to redefine it from what it was in Eq. 10. Please fix.

Supp Fig. 3 legend/12: "aligning with our expectations". Maybe I'm confused, but this seems silly; the Forward performance has to remain constant as a function of noise SD, so treating that as meeting an expectation is a tautology, no?

Supp 4/2/1: Please define the term "Forward(\sigma^2) agent" before you use it (and likewise for the other agents).

Supp Fig. 10 and elsewhere: Clarify that a pair of dots is a subject, or whatever, in the legend. The long sentence in the legend doesn't parse.

Figure 14: It seems silly to talk about "confidence noise with a standard deviation of 0" rather than to simply call is a noiseless model.

Reviewer #2: The manuscript presents a novel method for analyzing metacognitive efficiency in learning tasks. Traditional analysis methods are challenging to apply to these tasks, where choice difficulty changes dynamically. To address this, the authors have developed a measure called MetaRL.Ratio, analogous to the M-ratio measure　as it is defined by contrasting confidence-data-driven model performance with choice-data-driven model performance. This work has significant potential, broadening the scope of metacognitive studies to the learning domain, including future applications in the evaluation of non-human agent behavior. Below, I have several comments that would be considered for further improvement of the manuscript.

1. The proposed models exhibit worryingly high misfit to choice accuracy. This is concerning, as the validity of the proposed metacognitive measure is limited if the models do not successfully mimic the true generative process. The supplementary material reports the behavior of more complex models, but their choice accuracy does not appear to be presented. In any case, a more careful treatment of the misfit is expected.

2. On page 5, regarding the misfit of the backward model to choice accuracy, the authors state, “This was expected since the Backward model is fitted to make good predictions of confidence ratings, not choices.” I wonder if this is correct, since choice seems implicitly considered in equations 8 and 9 during the fitting of confidence data. Further methodological clarification around this point would be appreciated.

3. The present report includes the misfit of accuracy and the presumably undesirable correlation between MetaRL.Ratio and confidence bias. It seems the authors used the present simplified model as a provisional measurement model for performance evaluation. However, the proposed framework could also be applied to a more sophisticated process model, which is expected to better represent human data generation mechanisms. I would like the authors to offer a perspective or guideline on the applications they recommend.

4. The authors framed their approach analogously to meta-SDT. Yet, meta-SDT primarily concerns a posteriori data analysis, conducted while correct or incorrect responses are sorted out. In contrast, the proposed method seems to be conducted while trial-by-trial response correctness to be unknown, aligning more with a process model perspective (similar discussions can be found in Guggenmos, 2022, eLife). I would appreciate hearing the authors' insights on this matter.

**Have the authors made all data and (if applicable) computational code underlying the findings in their manuscript fully available?**

The PLOS Data policy requires authors to make all data and code underlying the findings described in their manuscript fully available without restriction, with rare exception (please refer to the Data Availability Statement in the manuscript PDF file). The data and code should be provided as part of the manuscript or its supporting information, or deposited to a public repository. For example, in addition to summary statistics, the data points behind means, medians and variance measures should be available. If there are restrictions on publicly sharing data or code —e.g. participant privacy or use of data from a third party—those must be specified.requires authors to make all data and code underlying the findings described in their manuscript fully available without restriction, with rare exception (please refer to the Data Availability Statement in the manuscript PDF file). The data and code should be provided as part of the manuscript or its supporting information, or deposited to a public repository. For example, in addition to summary statistics, the data points behind means, medians and variance measures should be available. If there are restrictions on publicly sharing data or code —e.g. participant privacy or use of data from a third party—those must be specified.requires authors to make all data and code underlying the findings described in their manuscript fully available without restriction, with rare exception (please refer to the Data Availability Statement in the manuscript PDF file). The data and code should be provided as part of the manuscript or its supporting information, or deposited to a public repository. For example, in addition to summary statistics, the data points behind means, medians and variance measures should be available. If there are restrictions on publicly sharing data or code —e.g. participant privacy or use of data from a third party—those must be specified.requires authors to make all data and code underlying the findings described in their manuscript fully available without restriction, with rare exception (please refer to the Data Availability Statement in the manuscript PDF file). The data and code should be provided as part of the manuscript or its supporting information, or deposited to a public repository. For example, in addition to summary statistics, the data points behind means, medians and variance measures should be available. If there are restrictions on publicly sharing data or code —e.g. participant privacy or use of data from a third party—those must be specified.

Reviewer #1: **No:** I don't see a link in the manuscript. Maybe I missed it...I don't see a link in the manuscript. Maybe I missed it...I don't see a link in the manuscript. Maybe I missed it...I don't see a link in the manuscript. Maybe I missed it...

Reviewer #2: **No:** Only the data are publicly accessible on their github without the codes.Only the data are publicly accessible on their github without the codes.Only the data are publicly accessible on their github without the codes.Only the data are publicly accessible on their github without the codes.

PLOS authors have the option to publish the peer review history of their article (what does this mean?). If published, this will include your full peer review and any attached files.). If published, this will include your full peer review and any attached files.). If published, this will include your full peer review and any attached files.). If published, this will include your full peer review and any attached files.

...

Reviewer #1: No

Reviewer #2: **Yes:** Kiyofumi MiyoshiKiyofumi MiyoshiKiyofumi MiyoshiKiyofumi Miyoshi

Figure Files:

While revising your submission, please upload your figure files to the Preflight Analysis and Conversion Engine (PACE) digital diagnostic tool, https://pacev2.apexcovantage.com. PACE helps ensure that figures meet PLOS requirements. To use PACE, you must first register as a user. Then, login and navigate to the UPLOAD tab, where you will find detailed instructions on how to use the tool. If you encounter any issues or have any questions when using PACE, please email us at . PACE helps ensure that figures meet PLOS requirements. To use PACE, you must first register as a user. Then, login and navigate to the UPLOAD tab, where you will find detailed instructions on how to use the tool. If you encounter any issues or have any questions when using PACE, please email us at . PACE helps ensure that figures meet PLOS requirements. To use PACE, you must first register as a user. Then, login and navigate to the UPLOAD tab, where you will find detailed instructions on how to use the tool. If you encounter any issues or have any questions when using PACE, please email us at . PACE helps ensure that figures meet PLOS requirements. To use PACE, you must first register as a user. Then, login and navigate to the UPLOAD tab, where you will find detailed instructions on how to use the tool. If you encounter any issues or have any questions when using PACE, please email us at figures@plos.org....
---

## [Decision Letter · Decision Letter 1]

30 Jul 2025

PCOMPBIOL-D-24-00793R1

Metacognitive Efficiency in Learned Value-based Choice

PLOS Computational Biology

Dear Dr. Ershadmanesh,

Thank you for submitting your manuscript to PLOS Computational Biology. After careful consideration, we feel that it has merit but does not fully meet PLOS Computational Biology's publication criteria as it currently stands. Therefore, we invite you to submit a revised version of the manuscript that addresses the points raised during the review process.

Please submit your revised manuscript within 30 days Sep 29 2025 11:59PM. If you will need more time than this to complete your revisions, please reply to this message or contact the journal office at ploscompbiol@plos.org. Please include the following items when submitting your revised manuscript:

We look forward to receiving your revised manuscript.

Kind regards,

Megan A. K. Peters, Ph.D.

Academic Editor

PLOS Computational Biology

Daniele Marinazzo

Section Editor

PLOS Computational Biology

**Additional Editor Comments :**

Please ensure you have confirmed all references to supplemental figures are correct, and ensure you are in compliance with the PLOS Data policy which requires authors to make all data and code underlying the findings described in their manuscript fully available without restriction, with rare exception.

**Journal Requirements:**

At this stage, the following Authors/Authors require contributions: Sara Ershadmanesh, Ali Gholamzadeh, Kobe Desender, and Peter Dayan. Please ensure that the full contributions of each author are acknowledged in the "Add/Edit/Remove Authors" section of our submission form.

2) Please ensure that the funders and grant numbers match between the Financial Disclosure field and the Funding Information tab in your submission form. Note that the funders must be provided in the same order in both places as well.

3) Regarding Figure 2A, please ensure that the full source details with direct link is included in the figure legend.

**Reviewers' comments:**

Reviewer's Responses to Questions

Reviewer #1: Re-review of "Metacognitive efficiency in learned value-based choice"

Reviewer: Michael Landy

Well, I still like this paper, and now I feel like I understood all of it. Most of my comments are fairly trivial. I'm pretty sure there are a bunch of Supplementary Figures that aren't cited in the main text, and the match of cited Supplementary Figure number is often wrong, so I suspect that the version of the Supplement that I got from clicking on the paper to download it might have been the wrong version of the Supplement. Anyway, here are my comments by line number (mostly):

118: Throughout my reading I was confused by what definition you had in mind for "confidence bias". After reading it I searched and found this parenthetic, and I think that's the definition you are using. Given the parameters that re-scale confidence, it's a wonder that anything about scaled mean confidence is all that meaningful. The scale parameters are constrained by the fits of the various models to the empirical confidence values and, in turn, are used to generate model confidence values. So, I'm not sure what it means when their averages don't match (a more or less skewed distribution over trials?). In any case, please highlight this as your definition of confidence bias.

168: "Figures 23 to 27" is a mismatch to the Supplement I have.

180: Here's the first use of the "confidence bias" term, so it needs to be preceded by your working definition.

189: Again, these seem like splitting hairs, as these differences are tiny compared to the range.

213: additions -> addition

Figure 3 legend, next to last line: estimations -> estimates

269: Another significant difference that seems insignificantly tiny

Table 1: Please get rid of scientific notation for all four of these p-values

325-326: grammar mess-up

330: Latex Equation-reference messup "??"

332: "was weaker" -> "was less well supported" or some such

335-336: "However, it did not fit better than". Well, it did fit better (the average AIC order is the same), just not "significnatly" based on your t-test of BIC values (which is kind of a weird thing to do, even though people often do it).

339: "Supplementary Figure 23" doesn't match my copy of the supplement

339-341: "were significantly closer to empirical performance" is followed by two tests that compare models to each other, not to the empirical performance. It's the next sentence that has tests that compare to empirical performance.

349: "was fitted" is kind of awkward phrasing

352: "Supplementary Figure 26" is another mismatch

449: les -> less

458/462: It's weird to have "a study" connected to two different citations.

513-514: This is a chi-squared goodness of fit to 50-50 where a reject means they were NOT guessing and should NOT be excluded. I asked for this text last time, but now see that you might as well have done an exact binomial test or the normal approximation for testing a proportion (I suppose the chi-squared is identical to that).

Eq. 6: You haven't said here how you map empirical confidence (ranging from 1 to 5) to [0,1] for this equation. Eq. 12 DOES do this scaling explicitly.

Eq. 18: The final term P(d | \theta) as stated should probably be the probability of a duration of d or greater. Alterantively, the initial P() should include d before the conditional.

546: You should define all the elements of the HMM parameters rather than assuming the reader knows what each one is. I figured most of them out from later equations. I never figured out what "n" represents.

Eq. 21 and line 553: It seems to me that if you want to know P(d|\theta), you are asking about a sequence that is, e.g., state 1, then state 2 for d trials, then state 1. If you condition on the first state, then the negative binomial you want is for n=1, so that n is not really a parameter.

Supplement: page/para/line

1/2/6: "our 56 empirical sequences". This is the first I've learned of a fixed set of sequences. One per subject? Where does this number come from?

Figure 4 legend: has a grammar mess-up (I tripped on "was a better model")

Figure 8 legend: "models" is missing an apostrophe somewhere

Figure 21B has "confidence bias" on the y-axis, so again a place that should follow a clear definition of how you compute confidence bias (as I say above)

Reviewer #2: The authors have successfully addressed all of my comments, and I would recommend the manuscript for publication.

**Have the authors made all data and (if applicable) computational code underlying the findings in their manuscript fully available?**

The PLOS Data policy requires authors to make all data and code underlying the findings described in their manuscript fully available without restriction, with rare exception (please refer to the Data Availability Statement in the manuscript PDF file). The data and code should be provided as part of the manuscript or its supporting information, or deposited to a public repository. For example, in addition to summary statistics, the data points behind means, medians and variance measures should be available. If there are restrictions on publicly sharing data or code —e.g. participant privacy or use of data from a third party—those must be specified.requires authors to make all data and code underlying the findings described in their manuscript fully available without restriction, with rare exception (please refer to the Data Availability Statement in the manuscript PDF file). The data and code should be provided as part of the manuscript or its supporting information, or deposited to a public repository. For example, in addition to summary statistics, the data points behind means, medians and variance measures should be available. If there are restrictions on publicly sharing data or code —e.g. participant privacy or use of data from a third party—those must be specified.requires authors to make all data and code underlying the findings described in their manuscript fully available without restriction, with rare exception (please refer to the Data Availability Statement in the manuscript PDF file). The data and code should be provided as part of the manuscript or its supporting information, or deposited to a public repository. For example, in addition to summary statistics, the data points behind means, medians and variance measures should be available. If there are restrictions on publicly sharing data or code —e.g. participant privacy or use of data from a third party—those must be specified.requires authors to make all data and code underlying the findings described in their manuscript fully available without restriction, with rare exception (please refer to the Data Availability Statement in the manuscript PDF file). The data and code should be provided as part of the manuscript or its supporting information, or deposited to a public repository. For example, in addition to summary statistics, the data points behind means, medians and variance measures should be available. If there are restrictions on publicly sharing data or code —e.g. participant privacy or use of data from a third party—those must be specified.

Reviewer #1: **No:** I don't see any reference to public availability of data. Maybe I missed it...I don't see any reference to public availability of data. Maybe I missed it...I don't see any reference to public availability of data. Maybe I missed it...I don't see any reference to public availability of data. Maybe I missed it...

Reviewer #2: **No:** Computational code seems to be missing on their github repository.Computational code seems to be missing on their github repository.Computational code seems to be missing on their github repository.Computational code seems to be missing on their github repository.

PLOS authors have the option to publish the peer review history of their article (what does this mean?). If published, this will include your full peer review and any attached files.). If published, this will include your full peer review and any attached files.). If published, this will include your full peer review and any attached files.). If published, this will include your full peer review and any attached files.

...

Reviewer #1: No

Reviewer #2: **Yes:** Kiyofumi MiyoshiKiyofumi MiyoshiKiyofumi MiyoshiKiyofumi Miyoshi

**Figure resubmission:**
---

## [Decision Letter · Decision Letter 2]

9 Mar 2026

Dear Dr. Ershadmanesh,

We are pleased to inform you that your manuscript 'Metacognitive Efficiency in Learned Value-based Choice' has been provisionally accepted for publication in PLOS Computational Biology.

Best regards,

Megan A. K. Peters, Ph.D.

Academic Editor

PLOS Computational Biology

Daniele Marinazzo

Section Editor

PLOS Computational Biology

Please attend to the Reviewer's comments and ensure the typo noted is fixed and figures in the supplemental material are high resolution.

Reviewer's Responses to Questions

**Comments to the Authors:**

Reviewer #1: One minor bug: Line 352: was less well SUPPORTED

Also, this time I wasn't given access to the supplement, so I couldn't check your responses to my comments there. The version of the manuscript I got did have some supplementary figures at the end, and the resolution of those images (they were TIFs rather than line art) was awful.

**Have the authors made all data and (if applicable) computational code underlying the findings in their manuscript fully available?**

The PLOS Data policy requires authors to make all data and code underlying the findings described in their manuscript fully available without restriction, with rare exception (please refer to the Data Availability Statement in the manuscript PDF file). The data and code should be provided as part of the manuscript or its supporting information, or deposited to a public repository. For example, in addition to summary statistics, the data points behind means, medians and variance measures should be available. If there are restrictions on publicly sharing data or code —e.g. participant privacy or use of data from a third party—those must be specified.requires authors to make all data and code underlying the findings described in their manuscript fully available without restriction, with rare exception (please refer to the Data Availability Statement in the manuscript PDF file). The data and code should be provided as part of the manuscript or its supporting information, or deposited to a public repository. For example, in addition to summary statistics, the data points behind means, medians and variance measures should be available. If there are restrictions on publicly sharing data or code —e.g. participant privacy or use of data from a third party—those must be specified.requires authors to make all data and code underlying the findings described in their manuscript fully available without restriction, with rare exception (please refer to the Data Availability Statement in the manuscript PDF file). The data and code should be provided as part of the manuscript or its supporting information, or deposited to a public repository. For example, in addition to summary statistics, the data points behind means, medians and variance measures should be available. If there are restrictions on publicly sharing data or code —e.g. participant privacy or use of data from a third party—those must be specified.requires authors to make all data and code underlying the findings described in their manuscript fully available without restriction, with rare exception (please refer to the Data Availability Statement in the manuscript PDF file). The data and code should be provided as part of the manuscript or its supporting information, or deposited to a public repository. For example, in addition to summary statistics, the data points behind means, medians and variance measures should be available. If there are restrictions on publicly sharing data or code —e.g. participant privacy or use of data from a third party—those must be specified.

Reviewer #1: Yes

PLOS authors have the option to publish the peer review history of their article (what does this mean?). If published, this will include your full peer review and any attached files.). If published, this will include your full peer review and any attached files.). If published, this will include your full peer review and any attached files.). If published, this will include your full peer review and any attached files.

...

Reviewer #1: **Yes:** Michael S. LandyMichael S. LandyMichael S. LandyMichael S. Landy

---

## [Editor Report · Acceptance letter]

PCOMPBIOL-D-24-00793R2

Metacognitive Efficiency in Learned Value-based Choice

Dear Dr Ershadmanesh,

I am pleased to inform you that your manuscript has been formally accepted for publication in PLOS Computational Biology. Your manuscript is now with our production department and you will be notified of the publication date in due course.

With kind regards,

Anita Estes
